# Online Differentially Private Conformal Prediction for Uncertainty Quantification

Qiangqiang Zhang [* 1]  Ting Li [* 2]  Xinwei Feng [1]  Xiaodong Yan [3]  Jinhan Xie [4]

## Abstract

Traditional conformal prediction faces significant challenges with the rise of streaming data and increasing concerns over privacy. In this paper, we introduce a novel online differentially private conformal prediction framework, designed to construct dynamic, model-free private prediction sets. Unlike existing approaches that either disregard privacy or require full access to the entire dataset, our proposed method ensures individual privacy with a one-pass algorithm, ideal for real-time, privacy-preserving decision-making. Theoretically, we establish guarantees for long-run coverage at the nominal confidence level. Moreover, we extend our method to conformal quantile regression, which is fully adaptive to heteroscedasticity. We validate the effectiveness and applicability of the proposed method through comprehensive simulations and real-world studies on the ELEC2 and PAMAP2 datasets.

## 1. Introduction

Conformal prediction (CP) is a powerful framework for quantifying predictive uncertainty in machine learning models. The fundamental principles of CP, meticulously detailed by Vovk et al. (2005), provided a rigorous basis for the construction of reliable prediction intervals, establishing CP as the cornerstone of uncertainty quantification. Building on these theoretical foundations, Balasubramanian et al. (2014) explored practical adaptations of CP, showcasing its effectiveness in diverse real-world applications and highlighting its potential to improve decision making in various domains.

Despite its broad applicability, the deployment of CP, particularly in domains that involve high-stakes decision-making, presents two significant challenges that must be addressed: 1. **Data Streaming and Concept Drift**: Data is received as continuous streams, accompanied by concept drift, the phenomenon where data distributions change over time, which can make traditional CP models, which rely on the exchangeability assumption, less effective. 2. **Data Privacy Concerns**: In domains such as healthcare and financial risk management, applying CP is often hampered by stringent privacy regulations. These sectors demand methods to provide accurate uncertainty quantification without compromising individual data privacy. Addressing these challenges is essential for the broader applicability of CP in real-world scenarios, especially when critical decisions are at stake and both data privacy and dynamic data environments must be taken into account.

In this paper, we introduce the online differentially private conformal prediction (ODPCP) framework, which leverages any pre-trained predictive model to construct dynamic differentially private prediction sets. Our framework integrates differential privacy mechanisms to ensure rigorous privacy protection while maintaining the reliability and adaptability of conformal prediction methods. In particular, we consider a scenario in which a data stream $\{(X_t, Y_t)\}_{1 \leq t \leq T}$ is generated by a dynamic process, allowing the distribution of $(X_t, Y_t)$ to evolve over time. At each time step $t$, our goal is to construct a dynamic differentially private prediction set for the target value $Y_t$ using previously observed data $\{(X_s, Y_s)\}_{s<t}$, and the newly observed covariates $X_t$. Our main **contributions** can be summarized as follows.

- **Online differentially private conformal prediction framework**: This paper presents a novel online differentially private conformal prediction framework that utilizes any pre-trained model to dynamically generate private prediction sets. In contrast to existing methods, our approach guarantees rigorous privacy protection without requiring access to the entire dataset, making it well-suited for real-time, privacy-preserving decision-making in an online manner.

- **One-pass algorithm**: Our proposed framework is

---
[*]Equal contribution [1]Zhongtai Securities Institute for Financial Studies, Shandong University, Jinan, China [2]School of Statistics and Data Science, Shanghai University of Finance and Economics, Shanghai, China [3]School of Mathematics and Statistics, Xi'an Jiaotong University, Xi'an, China [4]Yunnan Key Laboratory of Statistical Modeling and Data Analysis, Yunnan University, Kunming, China. Correspondence to: Jinhan Xie <jinhanxie@ynu.edu.cn>.

*Proceedings of the 42nd International Conference on Machine Learning*, Vancouver, Canada. PMLR 267, 2025. Copyright 2025 by the author(s).

an efficient one-pass algorithm that generates private prediction sets without the need to re-access historical data, thereby significantly reducing both time and space complexity.

- **Online differentially private conformal quantile prediction framework**: We extend our framework to conformal quantile regression (CQR), addressing heteroscedasticity by effectively managing data with varying levels of volatility. This extension improves both the accuracy and reliability of predictions in scenarios where the variability of data changes over time.

The structure of this paper is as follows: Section 2 reviews related work in this field. Section 3 introduces some key properties of differential privacy. In Section 4, we present the proposed online differentially private conformal prediction framework. Section 5 extends our framework to conformal quantile regression, addressing heteroscedasticity. Finally, in Section 6, we evaluate the performance of our method through some experiments, including simulations and two real-world case studies.

## 2. Related Work

CP has been widely extended to address numerous challenges, such as data distribution shifts and violations of exchangeability (Lei & Candès, 2021; Fannjiang et al., 2022; Barber et al., 2023; Plassier et al., 2024). Notably, Romano et al. (2019) and Kiyani et al. (2024) enhanced the reliability of prediction sets by optimizing the length of prediction intervals. Further developments in federated learning settings have been introduced by Lu et al. (2023) and Humbert et al. (2023), while Gasparin & Ramdas (2024a) and Gasparin & Ramdas (2024b) focused on conformal model aggregation, proposing novel strategies for efficiently merging uncertainty sets. Additional advancements in CP are discussed in works such as Bai et al. (2022), Liang et al. (2024), Xie et al. (2024), and Zecchin et al. (2024).

In recent years, significant advancements have been made in adapting CP frameworks to online settings, driven by the growing prevalence of data streams. Gibbs & Candes (2021) were pioneers in integrating online convex optimization techniques with CP frameworks. Building on this, subsequent work, such as Feldman et al. (2022); Bhatnagar et al. (2023), extended CP to support online environments, enabling real-time updates and adaptive prediction sets. For a more comprehensive overview of the latest developments in online CP, see, for example, (Angelopoulos et al., 2024b; Gibbs & Candès, 2024; Angelopoulos et al., 2024a). However, these methods generally lack built-in privacy-preserving mechanisms, leaving sensitive data susceptible to potential leakage and thereby limiting their applicability in privacy-sensitive contexts.

Differential Privacy (DP) has become a cornerstone in safeguarding privacy across a variety of sectors, including healthcare, information management, and government services; see Dankar & El Emam (2013); Wang & Tsai (2022). Prominent companies such as Google (Erlingsson et al., 2014), Microsoft (Ding et al., 2017), and the US Census Bureau (Garfinkel, 2022; Abowd, 2018) have adopted DP techniques to ensure the protection of individual privacy in large-scale systems. With the increasing global focus on privacy, the development of DP algorithms has expanded into diverse tasks, including, but not limited to, deep learning, demand learning, and transfer learning; see (Abadi et al., 2016; Bu et al., 2020; Chen et al., 2022; Ponomareva et al., 2023; Li et al., 2024). More recent advancements in the extension of DP can be found in works such as (Li et al., 2021; De et al., 2022; Nasr et al., 2023; Yang et al., 2024; Olabim et al., 2024).

The intersection of CP and DP has led to the development of privacy-preserving prediction sets. Early work, such as that by Angelopoulos et al. (2022), introduced algorithms that integrated the exponential mechanism (McSherry & Talwar, 2007) within the CP framework, ensuring privacy by obtaining private quantiles of non-conformity scores. Similarly, Humbert et al. (2023) extended this concept to one-shot federated learning, leveraging the exponential mechanism to guarantee privacy in federated settings. Plassier et al. (2023) took a different approach by adding noise to the gradient in order to derive private quantiles and generate DP prediction sets. While these approaches effectively ensure privacy, they are not directly applicable to online learning environments, where the need for real-time updates and continuous adaptation of prediction sets presents unique challenges.

## 3. Preliminary

In this section, we present basic concepts and some useful properties of DP.

**Definition 3.1** ((Dwork et al., 2006))**.** Let $\mathcal{X}$ be the sample space for an individual data, a randomized algorithm $A$: $\mathcal{X}^n \to \mathcal{R}$, is $(\epsilon, \delta)$-differentially private if and only if for every pair of adjacent datasets $X$ $X' \subset \mathcal{X}^n$ and for any measurable event $E \subseteq \mathcal{R}$, the inequality below holds:

$$\Pr(A(X) \in E) \le e^\epsilon \cdot \Pr(A(X') \in E) + \delta,$$

where we say that two datasets $X = \{x_i\}_{i=1}^n$ and $X' = \{x_i'\}_{i=1}^n$ are adjacent if and only if they differ by one individual datum and the probability measure $\Pr$ is induced by the randomness of $A$ only. When $\delta = 0$, then $M$ is called pure $\epsilon$-DP.

Specifically, when the dataset contains only a single data point, i.e., $|X| = 1$, the mechanism guarantees $(\epsilon, \delta)$-local differential privacy (LDP). This setting considers each data

point as a separate dataset, offering privacy protection at the individual level.

**Definition 3.2** ((Xiong et al., 2020)). A randomized algorithm $A : \mathcal{X} \to \mathcal{R}$ satisfies $(\epsilon, \delta)$-LDP if and only if for any pair of input individual values $x, x' \in \mathcal{X}$, and for every measurable event $E \subseteq \mathcal{R}$, the following inequality holds:

$$\Pr(A(x) \in E) \leq e^{\epsilon} \cdot \Pr(A(x') \in E) + \delta.$$

When $\delta = 0$, this reduces to pure $\epsilon$-LDP.

This property ensures that our framework offers privacy protection at the individual level, safeguarding sensitive information even when the data consists of just one observation. Many fundamental algorithms can be rendered DP by adding noise, appropriately scaled according to the algorithm's sensitivity. The formal definition of sensitivity is provided as follows:

**Definition 3.3** ($\ell_p$-sensitivity). Let $h : \mathcal{X}^n \to \mathcal{R}^d$ be a query mapping. For a fixed positive scalar $p$, the $\ell_p$-sensitivity of $h$ is defined by $\Delta_p(h) = \sup_{X, X' \subset \mathcal{X}^n, adjacent} \|h(X) - h(X')\|_p$.

Building upon the concept of sensitivity, we next introduce Gaussian Differential Privacy (GDP), a more recent formulation that captures privacy guarantees through hypothesis testing between adjacent datasets.

**Definition 3.4** ((Dong et al., 2022)). Let $\mathcal{X}$ be the sample space and let $A : \mathcal{X}^n \to \mathcal{R}$ be a randomized algorithm. For all adjacent datasets $X, X' \in \mathcal{X}^n$, we say that $A$ satisfies $\mu$-Gaussian Differential Privacy (abbreviated as $\mu$-GDP) if the trade-off function between $A(X)$ and $A(X')$ is pointwise lower bounded by that of $\mathcal{N}(0, 1)$ and $\mathcal{N}(\mu, 1)$, that is,

$$T \left( A(X), A(X') \right) (\alpha) \geq G_\mu(\alpha), \quad \forall \alpha \in [0, 1].$$

For any two probability distributions $P$ and $Q$ on the same measurable space, the trade-off function $T(P, Q) : [0, 1] \to [0, 1]$ is defined as

$$T(P, Q)(\alpha) := \inf \{\beta_\phi : \alpha_\phi \leq \alpha\},$$

where $\phi : \mathcal{R} \to [0, 1]$ is a measurable rejection rule, and $\alpha_\phi$ and $\beta_\phi$ denote the type I and type II errors, respectively. This function quantifies the minimum type II error achievable under a level-$\alpha$ type I error constraint.

The Gaussian trade-off function $G_\mu$ is given by

$$G_\mu(\alpha) := \Phi \left( \Phi^{-1}(1 - \alpha) - \mu \right),$$

where $\Phi(\cdot)$ is the standard normal CDF and $\Phi^{-1}(\cdot)$ is its inverse.

The following are commonly used mechanisms that enable the design of our proposed privacy-preserving algorithm.

**Lemma 3.5.** *(Dwork et al., 2014)*

1. ***Laplace Mechanism***: *Let $h(X) \in \mathcal{R}^d$ be a statistic of the dataset $X$, with a finite $\ell_1$-sensitivity $\Delta_1(h)$. The Laplace mechanism adds noise to the output of $h$ as follows:*

$$M(X) = h(X) + \omega,$$

   *where each component $\omega_1, \ldots, \omega_d$ of $\omega \in \mathcal{R}^d$ is independently drawn from the Laplace distribution $Laplace(0, \Delta_1(h)/\epsilon)$. The output of this mechanism $M$ satisfies $\epsilon$-DP.*

2. ***Gaussian Mechanism***: *Let $h(X) \in \mathcal{R}^d$ represent a statistic derived from the dataset $X$, with $\ell_2$-sensitivity $\Delta_2(h)$. The Gaussian mechanism operates by adding noise to the output of $h$ in the following manner:*

$$M(X) = h(X) + \omega,$$

   *where each component $\omega_1, \ldots, \omega_d$ of $\omega \in \mathcal{R}^d$ is drawn independently from the distribution $N(0, 2(\Delta_2(h)/\epsilon)^2 \log(1.25/\delta))$. The output of this mechanism $M$ satisfies $(\epsilon, \delta)$-DP.*

3. ***$\mu$-GDP***: *Let $h(X) \in \mathcal{R}^d$ be a statistic computed from the dataset $X$, with $\ell_2$-sensitivity $\Delta_2(h)$. The Gaussian Differential Privacy (GDP) mechanism applies noise to $h$ according to the following:*

$$M(X) = h(X) + \omega,$$

   *where each component $\omega_1, \ldots, \omega_d$ of $\omega \in \mathcal{R}^d$ is independently sampled from $N \left(0, (\Delta_2(h)/\mu)^2\right)$. The output of this mechanism $M$ satisfies $\mu$-GDP.*

After presenting the concept of differential privacy and its underlying mechanisms, we now proceed to outline key results that demonstrate how privacy guarantees are preserved across various operations.

**Lemma 3.6.** *(Dwork et al., 2006; 2010)*

***Post-processing property***: *Let $(Z, \mathcal{Z})$ be a measurable space, and let $A : X^n \to Y$ be an $(\epsilon, \delta)$-DP. If $f : Y \to Z$ is a measurable function, then the composition $f \circ A$ is also $(\epsilon, \delta)$-DP.*

***Sequential composition***: *Suppose there are mechanisms $A_i : X^n \to Y$ for $i = 1, \ldots, k$, where each mechanism satisfies $(\epsilon_i, \delta_i)$-DP. Then the combined mechanism $A : X^n \to Y^k$, which maps $x$ to $(A_1(x), \ldots, A_k(x))$, satisfies $(\sum_{i=1}^{k} \epsilon_i, \sum_{i=1}^{k} \delta_i)$-DP.*

***Parallel composition***: *Let $A_i : X^n \to Y$ for $i = 1, \ldots, k$ be mechanisms that each satisfy $(\epsilon_i, \delta_i)$-DP. If the datasets $x_1, \ldots, x_k \in X^n$ are disjoint, then the combined mechanism $A : X^{n \times k} \to Y^k$, which maps $(x_1, \ldots, x_k)$ to $(A_1(x_1), \ldots, A_k(x_k))$, satisfies $(\max_i \epsilon_i, \max_i \delta_i)$-DP.*

# 4. Online differentially private conformal prediction

Recall that CP constructs prediction regions based on exchangeable data $\{(X_i, Y_i)\}_{i=1}^n$ and a pre-trained model $\hat{f}$. The fundamental idea behind CP is to quantify prediction uncertainty through **non-conformity scores** $S(X, Y) = S_{\hat{f}}(X, Y)$, which capture the discrepancy between observations and model predictions (e.g., residuals in regression tasks or prediction confidence in classification tasks). By computing a quantile threshold from the calibration set, CP generates prediction regions that ensure the desired coverage probability, formally expressed as:

$$\mathbb{P}\big(Y_{\text{new}} \in C(X_{\text{new}})\big) \geq 1 - \alpha,$$

providing a statistically valid characterization of model uncertainty. Here, $1 - \alpha$ is the target nominal level, and $(X_{\text{new}}, Y_{\text{new}})$ denotes the new observation.

However, when applied to data streams $\{(X_t, Y_t)\}_{t \geq 1}$, the traditional CP framework faces several challenges. First, temporal distribution shifts undermine the exchangeability assumption, which is vital for maintaining valid coverage. Second, static calibration methods fail to adapt to the dynamic nature of evolving data patterns. Third, the batch quantile computation required by CP becomes computationally expensive and impractical for real-time applications. In addition to these technical challenges, privacy concerns have become increasingly critical in modern applications such as healthcare and finance.

To address these challenges, we propose the ODPCP framework. ODPCP dynamically constructs prediction sets that adapt to streaming data while ensuring formal differential privacy guarantees. In contrast to conventional methods, our framework enforces privacy by perturbing the quantile estimation process itself. Specifically, we estimate the $(1 - \alpha)$-quantile of non-conformity scores $\{S_t\}$ using subgradient-based updates, injecting noise into each subgradient to ensure differential privacy.

At each time step $t$, we estimate the $(1 - \alpha)$-quantile $\hat{q}_t^{1-\alpha}$ of the non-conformity scores $S_t$ by minimizing the pinball loss:

$$\ell_{1-\alpha}(q, S_t) = (\mathbf{1}\{q \geq S_t\} - (1 - \alpha))(q - S_t) \quad (1)$$

which has Lipschitz constant $L = \max\{\alpha, 1 - \alpha\}$. The corresponding subdifferential is:

$$\partial \ell_{1-\alpha}(q, S_t) = \begin{cases} \mathbf{1}\{S_t \leq q\} - (1 - \alpha), & q \neq S_t \\ [\alpha - 1, \alpha], & q = S_t \end{cases} \quad (2)$$

This yields a subgradient update:

$$g_t \in \partial \ell_{1-\alpha}(q, S_t)\big|_{q=\hat{q}_t^{1-\alpha}}. \quad (3)$$

To ensure privacy, we define the privatized subgradient:

$$\hat{g}_t = g_t + \mathcal{Z}_t, \quad \mathcal{Z}_t \sim \text{i.i.d. noise.} \quad (4)$$

This mechanism ensures that each update step satisfies differential privacy. A conventional approach would apply stochastic gradient descent (SGD):

$$\hat{q}_{t+1}^{1-\alpha} = \hat{q}_t^{1-\alpha} - \eta_t g_t, \quad (5)$$

but this requires manual tuning of the learning rate $\eta_t$, which becomes especially problematic in the presence of noise, as introduced by $\hat{g}_t$. The noise can amplify instability, degrade convergence speed, and complicate hyperparameter selection. To avoid these challenges, we draw inspiration from the coin-betting framework in parameter-free online convex optimization (Orabona & Pál, 2016; Cutkosky & Orabona, 2018; Podkopaev et al., 2024), which enables adaptive learning without manual tuning of step sizes.

*Remark* 4.1. Our framework incorporates various mechanisms to ensure privacy protection. For instance, we can add noise to the gradients in the following ways: 1. **Laplace mechanism:** $\mathcal{Z}_t \sim$ Laplace$(0, \Delta_1(\partial \ell_{1-\alpha})/\epsilon_t)$; 2. **Gaussian mechanism:** $\mathcal{Z}_t \sim N(0, 2(\Delta_2(\partial \ell_{1-\alpha})/\epsilon_t)^2 \log(1.25/\delta_t))$; 3. $\mu_t$-**GDP:** $\mathcal{Z}_t \sim N(0, (\Delta_2(\partial \ell_{1-\alpha})/\mu_t)^2)$. From the above-perturbed formula in (4), we observe that each individual has the flexibility to customize its privacy budget. A special case happens when we use a uniform privacy budget without considering different privacy budgets for each individual. Throughout this paper, we refer to our algorithm as DP. Nonetheless, its implementation aligns with the principles of LDP, as privacy is enforced via local perturbation at each step rather than through a centralized mechanism.

This parameter-free and robust approach reformulates the learning process as a repeated betting game. In this framework, a bettor wagers a fraction $\lambda_t$ of their current wealth $W_{t-1}$ on an outcome $c_t$, which may be adversarially selected. Starting from an initial endowment $W_0 > 0$, the cumulative wealth evolves as $W_t = W_0 + \sum_{i=1}^t \lambda_i W_{i-1} c_i$. In our setup, to integrate this framework with privatized quantile estimation, we define the feedback signal as $c_t := -\hat{g}_t$, where $\hat{g}_t$ denotes the privatized subgradient. We then employ the Krichevsky–Trofimov (KT) estimator (Krichevsky & Trofimov, 1981), which sets the betting fraction as $\lambda_t = \sum_{i=1}^{t-1} c_i / t = -\sum_{i=1}^{t-1} \hat{g}_i / t$. The signed bet $w_t := \lambda_t W_{t-1}$ serves as the current estimate of the privatized $(1 - \alpha)$-quantile, denoted $\hat{q}_t^{1-\alpha}$.

In this setup, we introduce two crucial parameters: the cumulative wealth $W_t$ and the betting ratio $\lambda_t$, both of which are adaptively adjusted to balance exploration and exploitation throughout the process. Our proposed algorithm eliminates the need for manual hyperparameter tuning by automatically adjusting these two parameters, while also integrating DP

---

**Algorithm 1** Online private conformal prediction set

---
**Input:** Data $(X_t, Y_t), t = 1, 2, \ldots$; Prediction model $\hat{f}_t(\cdot)$; Privacy budget parameters $\epsilon_t > 0$; $\alpha \in (0, 1)$; Constant $c > 0$; $S_1 > 0$
**for** $t = 2, 3, \ldots$ **do**
  Estimate the privatized quantile :
  $\hat{q}_t^{1-\alpha} \leftarrow$ Algorithm 2($\epsilon_{t-1}, \alpha, S_{t-1}, c$);
  Predict and compute the non-conformity score $S_t = S_{\hat{f}_t}(X_t, Y_t)$.
**end for**
**Output:**
Private prediction sets $\hat{C}_t(X_t) = \{y : S_{\hat{f}_t}(X_t, y) \leq \hat{q}_t^{1-\alpha}\}$

---

**Algorithm 2** Online differentially private quantile

---
**Input:** Privacy budget parameter $\epsilon_{t-1} > 0$; Miscoverage level $\alpha \in (0, 1)$; $S_{t-1}$; $c > 0$
// Initialize parameters (only used on the first call, i.e., when $t = 2$)
Initialize: $W_0 = 1, \lambda_1 = 0, \hat{q}_1^{1-\alpha} = 0$
Compute $g_{t-1} \in \partial \ell_{1-\alpha}(q, S_{t-1})|_{q=\hat{q}_{t-1}^{1-\alpha}}$ as per Equation (2)
Perturb the gradient: $\hat{g}_{t-1} = g_{t-1} + \mathcal{Z}_{t-1}$, where $\mathcal{Z}_{t-1}$ is the random noise according to different DP mechanisms
Update $W_{t-1}$:

$$W_{t-1} = (W_{t-2} - \hat{g}_{t-1}\hat{q}_{t-1}^{1-\alpha}) \vee c$$

Update $\lambda_t$:

$$\lambda_t = \frac{t-1}{t}\lambda_{t-1} - \frac{1}{t}\hat{g}_{t-1}$$

Update $\hat{q}_t^{1-\alpha}$:
$$\hat{q}_t^{1-\alpha} = \lambda_t W_{t-1}$$

Save updated state: $W_{t-1}, \lambda_t, \hat{q}_t^{1-\alpha}$

---

mechanisms. This design ensures both statistical validity and privacy protection of the quantile estimates, allowing the method to efficiently navigate the trade-off between exploration and exploitation, while maintaining privacy guarantees.

To ensure stability in the learning process under the influence of noise, we introduce a constant $c$ as a lower bound for the parameter $W_t$. This lower bound prevents $W_t$ from becoming too small or negative due to noise perturbations, which are particularly impactful during the early iterations. By enforcing this constraint, we facilitate a smooth initialization phase, mitigate the disruptive effects of noise, and ultimately support the reliable convergence of the algorithm.

*Remark* 4.2. We conduct a sensitivity analysis for the proposed framework regarding the constant $c$. As shown in Figure 1, we observe that, as $c$ increases, the coverage rate stabilizes, while the interval length grows more rapidly. For smaller values of $c$, increasing $c$ results in a moderate widening of the interval and an improvement in coverage, enhancing the practical effectiveness of the prediction set. However, for larger values of $c$, further increases in interval length yield diminishing returns in terms of coverage improvement, and may even reduce the information content of the interval. Based on these findings, we recommend choosing $c$ within the range of 30 to 50, as this range strikes a trade-off balance between improving coverage and maintaining the interval's informational value.

We present the pseudocode of our proposed method in Algorithms 1–2. Algorithm 1 computes the non-conformity scores, while Algorithm 2 dynamically updates the quantile estimates in a privacy-preserving manner. Together, these algorithms ensure that the conformal prediction sets maintain valid coverage guarantees under differential privacy. The final output is a private prediction set $\hat{C}_t(X_t)$ that effectively balances privacy protection and predictive reliability.

*Remark* 4.3. When the predictive model is trained with DP, the entire pipeline—from the data to the prediction set

function $\hat{C}_t(\cdot)$—becomes DP. To assess the impact of this, we compare the algorithm's performance using both non-private and DP-trained base models. The corresponding results are reported in Appendix C.1.2 and Appendix C.2.3.

**Theorem 4.4.** *After* $t$ *updates, Algorithm 1 satisfies* $\max_{1 \leq j \leq t} \epsilon_j$*-DP,* $(\max_{1 \leq j \leq t} \epsilon_j, \max_{1 \leq j \leq t} \delta_j)$*-DP, or* $\max_{1 \leq j \leq t} \mu_j$*-GDP, depending on the privacy mechanism used, where* $t \geq 1$.

This theorem rigorously establishes the privacy guarantees of Algorithm 1 under different DP mechanisms. Our experimental results align well with the theoretical guarantees stated in Theorem 4.4, as demonstrated in Appendix C.2.6.

**Theorem 4.5.** *Let the target miscoverage level be fixed at* $\alpha \in (0, 1/2)$. *Assume that the non-conformity scores are bounded, such that* $S_t \in [0, D]$ *for all* $t = 1, 2, \ldots$, *where* $D > 0$ *is a finite constant. Then, the proposed procedure described in Algorithm 1 satisfies the following long-run coverage guarantee:*

$$\lim_{T \to \infty} \left| \frac{1}{T} \sum_{t=1}^{T} \mathbb{I}\left(Y_t \in \hat{C}_t(X_t)\right) - (1-\alpha) \right| = 0. \quad (6)$$

Theorem 4.5 establishes that the proposed method ensures valid long-run coverage guarantees, regardless of the underlying predictive model and whether privacy constraints are present. However, privately trained models often suffer reduced predictive accuracy due to injected noise. Additional results in Appendix C.1.2 and Appendix C.2.3 show that

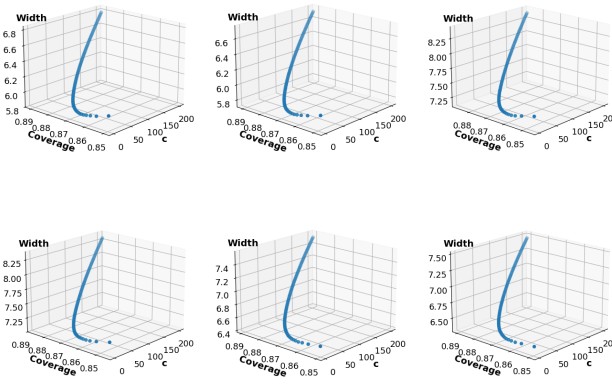

*Figure 1.* Sensitivity analysis of different values of $c$ on the coverage rate and interval length under the $\mu$-GDP setting. The constant $c$ is varied from 1 to 200 in steps of 2. For clarity, the results presented here are based on the Setting 2 dataset. The figures are arranged from left to right, corresponding to Case 1 through Case 6. Each simulation is run for $T = 10,000$ time steps.

such models typically yield larger non-conformity scores, leading to wider prediction intervals.

## 5. Online differentially private conformal quantile regression

In this section, we extend our framework to incorporate CQR (Romano et al., 2019), enabling our method to effectively address heteroscedasticity and improve its adaptability to varying data volatility.

To begin, we recall the concept of the conditional distribution function, $F(y|X = x)$, which represents the probability that $Y$ is less than or equal to a certain value $y$ given that $X = x$. The corresponding $\alpha$-th conditional quantile function is denoted as $q_\alpha(x)$. To construct the prediction set, we define two quantiles: $\alpha_{\text{lo}} = \alpha/2$ and $\alpha_{\text{hi}} = 1 - \alpha/2$, where $\alpha \in (0, 1)$ represents the miscoverage level. The prediction interval is then given as follows:

$$C(x) = [q_{\alpha_{\text{lo}}}(x), q_{\alpha_{\text{hi}}}(x)]$$

We estimate $\hat{q}_{\alpha_{\text{lo}}}(x)$ and $\hat{q}_{\alpha_{\text{hi}}}(x)$ using quantile regression and the non-conformity score here is defined as

$$S_t := \max\left\{\hat{q}_{\alpha_{\text{lo}}}(X_t) - Y_t, Y_t - \hat{q}_{\alpha_{\text{hi}}}(X_t)\right\}$$

for each time point $t$, which captures the discrepancy between the observed target value and the predicted quantiles, reflecting how far the observation deviates from the expected range. Using this non-conformity score, the prediction interval for $Y_t$ at time $t$ is constructed as:

$$\hat{C}_t(X_t) = \left[\hat{q}_{\alpha_{\text{lo}}}(X_t) - \hat{q}_t^{1-\alpha}, \hat{q}_{\alpha_{\text{hi}}}(X_t) + \hat{q}_t^{1-\alpha}\right].$$

We continue to compute $g_t$ using $\partial\ell_{1-\alpha}(q, S_t)$, preserving the iterative structure of the algorithm. This approach dynamically adjusts the length of the prediction interval based on the behavior of the non-conformity score $S_t$. When $S_t$ is less than $\hat{q}_t^{1-\alpha}$, there are two possible cases: If $S_t$ is negative, it indicates that the observed value $Y_t$ falls within the prediction interval $\hat{C}_t(X_t)$. If $S_t$ is positive, it suggests that $Y_t$ lies either to the left of $\hat{q}_{\alpha_{\text{lo}}}(X_t)$ or to the right of $\hat{q}_{\alpha_{\text{hi}}}(X_t)$, but the distance between $Y_t$ and the nearest endpoint of the interval remains less than $\hat{q}_t^{1-\alpha}$. In both cases, it follows that $Y_t \in \hat{C}_t(X_t)$. Consequently, the value of $g_t$ will be positive, and the prediction interval will be moderately reduced. Conversely, when $S_t$ exceeds $\hat{q}_t^{1-\alpha}$, $Y_t$ lies outside the prediction interval, either to the left of $\hat{q}_{\alpha_{\text{lo}}}(X_t)$ or to the right of $\hat{q}_{\alpha_{\text{hi}}}(X_t)$. This implies that the distance between $Y_t$ and the nearest endpoint exceeds $\hat{q}_t^{1-\alpha}$, meaning $Y_t \notin \hat{C}_t(X_t)$. In this case, the value of $g_t$ will be negative, and the algorithm will appropriately expand the interval width.

This process allows our algorithm to adapt the length of the prediction interval in real-time, ensuring that it is both accurate and efficient. By dynamically adjusting the prediction set based on the behavior of $S_t$, the algorithm maintains its reliability and convergence, even in the presence of evolving data streams. This ensures that our method can provide correct predictions while simultaneously preserving privacy guarantees, making it suitable for applications in dynamic and privacy-sensitive environments. The details of this algorithm can be summarized in Algorithm 3 in the Appendix.

## 6. Experiments

In this section, we present an empirical evaluation of our proposed method, comparing its performance with a non-private baseline (Podkopaev et al., 2024), referred to as "Original". This experimental setup consists of two components. First, we conduct simulations on synthetic data to evaluate the performance of the proposed method. Second, we assess the method's effectiveness on real-world datasets.

Notice that the main contribution of this paper is the development of a privacy-preserving method based on the conformal prediction framework. Our focus is on how to incorporate privacy protection into the conformal prediction setting, rather than on designing or training the private predictor itself. Therefore, in the subsequent experiments, we do not treat the private predictor as a necessary condition, allowing us to isolate and highlight the core functionality of the proposed method. For conciseness, the main text focuses on the results obtained using the $\mu$-GDP mechanism (via Gaussian noise), while the results of using $\epsilon$-DP with Laplace noise, as well as detailed experiments on setting 1 and the ELEC2 dataset, are provided in the Appendix.

## 6.1. Simulations

We evaluate performance via the following two data generation settings:

**Setting 1**: We generate $N = 10,000$ samples $(X_t, Y_t)$,

$$Y_t = X_t^\top \beta + \varepsilon_t, \quad \beta = (1, 0.5, 1, 0, 0)^\top,$$

and consider two cases:

- **Case 1:** $X_t \overset{\text{iid}}{\sim} \mathcal{N}(0, I_5), \varepsilon_t \overset{\text{iid}}{\sim} \mathcal{N}(0, 1)$;

- **Case 2:** $X_t \overset{\text{iid}}{\sim} \mathcal{N}(0, \Sigma)$, where $\Sigma_{ij} = 0.5^{|i-j|}, \varepsilon_t \overset{\text{iid}}{\sim} \mathcal{N}(0, 1)$.

**Setting 2**: Similar to Setting 1, where the coefficients $\beta^{(t)}$ change at $t = 2500$ and $t = 7500$:

$$\beta^{(1:2500)} = (1, 0.5, 1, 0, 0)^\top,$$
$$\beta^{(2501:7500)} = (0, -1, -0.5, -1, 0)^\top,$$
$$\beta^{(7501:10000)} = (0, 0, 1, 0.5, 1)^\top.$$

We also consider the following cases:

- **Case 1:** $X_t \overset{\text{iid}}{\sim} \mathcal{N}(0, I_5), \varepsilon_t \overset{\text{iid}}{\sim} \mathcal{N}(0, 1)$;

- **Case 2:** $X_t \overset{\text{iid}}{\sim} \mathcal{N}(0, \Sigma)$, where $\Sigma_{ij} = 0.5^{|i-j|}, \varepsilon_t \overset{\text{iid}}{\sim} \mathcal{N}(0, 1)$;

- **Case 3:** $X_t \overset{\text{iid}}{\sim} \mathcal{N}(0, I_5), \varepsilon_t \overset{\text{iid}}{\sim} \mathsf{t}(3)$;

- **Case 4:** $X_t \overset{\text{iid}}{\sim} \mathcal{N}(0, \Sigma)$, where $\Sigma_{ij} = 0.5^{|i-j|}, \varepsilon_t \overset{\text{iid}}{\sim} \mathsf{t}(3)$;

- **Case 5:** $X_t \overset{\text{iid}}{\sim} \mathcal{N}(0, I_5), \varepsilon_t = X_{t1}^2 \cdot \zeta_t$, where $\zeta_t \overset{\text{iid}}{\sim} \mathcal{N}(0, 1)$;

- **Case 6:** $X_t \overset{\text{iid}}{\sim} \mathcal{N}(0, \Sigma)$, where $\Sigma_{ij} = 0.5^{|i-j|}, \varepsilon_t = X_{t1}^2 \cdot \zeta_t$, where $\zeta_t \overset{\text{iid}}{\sim} \mathcal{N}(0, 1)$.

These cases are designed to systematically evaluate the robustness of our method along three key dimensions: (i) covariate structure (independent vs. correlated designs), (ii) noise distribution (Gaussian, heavy-tailed, and heteroscedastic), and (iii) temporal dynamics (static vs. time-varying coefficients). Setting 1 corresponds to a static environment with fixed regression coefficients, while Setting 2 introduces changepoints to emulate dynamic, nonstationary conditions. This design ensures a comprehensive and realistic assessment of our method under a wide range of practical scenarios.

As shown in Tables 1–2 and Figure 2 and Figure 14, both the proposed ODPCP method and ODPCQR demonstrate consistent and reliable performance across the six different simulated scenarios. Specifically, both methods exhibit stable

*Table 1.* Long-Run Coverage $\sum_{t=1}^{T} \mathbb{I}\{Y_t \in \hat{C}_t(X_t)\}/T$, and the standard deviation (scaled by a factor of 1,000), are reported for the Setting 2 dataset with $\mu$-GDP, averaged over 200 independent trials . To minimize the impact of early-stage noise and the initial instability of the algorithm, the first 100 data points are excluded from the analysis. **C** represents cases and **M** represents methods.

| C/M | Original | $\mu = 2$ | $\mu = 1$ | $\mu = 0.5$ |
|---|---|---|---|---|
| 1/CP | 0.889 (0.29) | 0.886 (5.3) | 0.874 (10) | 0.850 (19) |
| 1/CQR | 0.890 (1.2) | 0.895 (4.5) | 0.895 (8.9) | 0.895 (18) |
| 2/CP | 0.888 (0.28) | 0.886 (4.7) | 0.874 (9.4) | 0.841 (17) |
| 2/CQR | 0.891 (1.4) | 0.895 (4.4) | 0.895 (8.4) | 0.896 (17) |
| 3/CP | 0.889 (0.32) | 0.887 (5.3) | 0.874 (10) | 0.846 (18) |
| 3/CQR | 0.892 (1.6) | 0.897 (4.4) | 0.897 (8.6) | 0.898 (17) |
| 4/CP | 0.888 (0.33) | 0.886 (5.2) | 0.872 (10) | 0.834 (18) |
| 4/CQR | 0.893 (1.5) | 0.896 (4.4) | 0.895 (8.3) | 0.895 (17) |
| 5/CP | 0.889 (0.37) | 0.886 (5.3) | 0.874 (10) | 0.848 (18) |
| 5/CQR | 0.892 (1.7) | 0.896 (4.2) | 0.896 (8.2) | 0.898 (17) |
| 6/CP | 0.888 (0.35) | 0.886 (4.9) | 0.874 (9.6) | 0.839 (17) |
| 6/CQR | 0.891 (1.8) | 0.895 (4.2) | 0.895 (8.0) | 0.895 (16) |

coverage rates and appropriately scaled prediction interval lengths, underscoring their adaptability and effectiveness in tackling complex structures. These results highlight the robustness of both algorithms in providing privacy-preserving, reliable prediction intervals under challenging conditions.

Figure 2 and Figure 14 illustrate that the trends of the proposed private algorithms closely mirror those of the Original algorithm. Specifically, when the privacy parameter $\mu$ is large (indicating weaker privacy protection), the performance of the private algorithms becomes nearly indistinguishable from that of the Original algorithm, with coverage rates and prediction interval lengths aligning closely. This suggests that as the noise introduced by the privacy mechanism diminishes, its effect on the algorithm's performance becomes negligible. Even with stronger privacy protection (e.g., $\mu = 0.5$), the private algorithms maintain a similar trend to the Original algorithm, albeit with greater variability in coverage and interval width, indicating that a balance between privacy and prediction accuracy is being struck. Meanwhile, at changepoints, both the coverage rates and prediction interval lengths experience significant fluctuations, with coverage rates sharply decreasing and interval lengths expanding considerably. Despite these disruptions, both algorithms promptly stabilize, demonstrating their resilience. As the value of $\mu$ decreases (indicating stronger privacy protection), the coverage rates of our proposed method decrease, and the prediction interval lengths become larger, accompanied by more pronounced fluctuations. This underscores the growing effect of noise on the algorithm's stability as the level of privacy protection increases.

In scenarios characterized by high noise and strong privacy

*Table 2.* Long-Run Width and standard deviation (scaled by 10) are reported for the Setting 2 dataset with $\mu$-GDP, consistent with Table 1.

| C/M | Original | $\mu = 2$ | $\mu = 1$ | $\mu = 0.5$ |
|---|---|---|---|---|
| 1/CP | 5.18 (0.45) | 5.23 (0.95) | 5.46 (2.5) | 7.69 (23) |
| 1/CQR | 5.11 (0.46) | 5.18 (0.85) | 5.25 (1.7) | 5.90 (12) |
| 2/CP | 6.49 (0.60) | 6.56 (1.1) | 6.84 (3.1) | 9.68 (33) |
| 2/CQR | 6.39 (0.68) | 6.48 (1.0) | 6.55 (1.9) | 7.24 (13) |
| 3/CP | 6.22 (0.60) | 6.32 (1.4) | 6.67 (3.8) | 9.93 (40) |
| 3/CQR | 6.22 (0.72) | 6.33(1.3) | 6.42 (2.4) | 7.08 (10) |
| 4/CP | 7.45 (0.77) | 7.53 (1.5) | 7.88 (3.8) | 11.48 (58) |
| 4/CQR | 7.40 (0.79) | 7.48 (1.3) | 7.55 (2.4) | 8.28 (27) |
| 5/CP | 5.81 (0.87) | 5.90 (1.8) | 6.21 (4.5) | 8.95 (61) |
| 5/CQR | 5.85 (1.1) | 5.96 (1.6) | 6.05 (2.8) | 7.08 (10) |
| 6/CP | 7.04 (0.87) | 7.13 (1.9) | 7.47 (4.6) | 10.26 (30) |
| 6/CQR | 7.04 (1.1) | 7.16 (1.6) | 7.24 (2.9) | 7.86 (9.8) |

protection, the proposed ODPCQR demonstrates superior performance compared to ODPCP, exhibiting both higher coverage rates and shorter prediction intervals. This indicates that ODPCQR is more adaptable to changes in $\mu$. Notably, the coverage rates of the proposed ODPCQR between the two changepoints slightly exceed the target coverage rate of 0.9. Furthermore, its coverage even surpasses that of the non-private algorithm. This improvement can be attributed to the compensatory effect of the larger prediction intervals induced by the added noise, which helps to ensure better coverage. In conclusion, the simulation results affirm that both algorithms deliver consistent and reliable performance in the complex data environment of Setting 2. They exhibit remarkable adaptability and robustness, effectively handling a variety of challenging scenarios. These outcomes highlight the capacity of the proposed methods to maintain performance even under conditions of data variability and noise.

Moreover, our methods exhibit negligible sensitivity to the choice of initial parameters $(\lambda_1, W_0)$, as demonstrated in Figure 16, where the coverage and interval width curves under different initializations are nearly indistinguishable. In addition, we compare our method with an offline private baseline—Private Prediction Sets (DPCP) (Angelopoulos et al., 2022). As shown in Table 11, both methods achieve comparable coverage. However, ODPCP produces significantly narrower prediction sets, highlighting its efficiency in the online setting under privacy constraints.

### 6.2. PAMAP2 physical activity monitoring dataset

In this subsection, we assess the performance of the proposed method using the PAMAP2 physical activity monitoring dataset (Reiss, 2012). This dataset includes measurements from 9 subjects engaged in various physical activities,

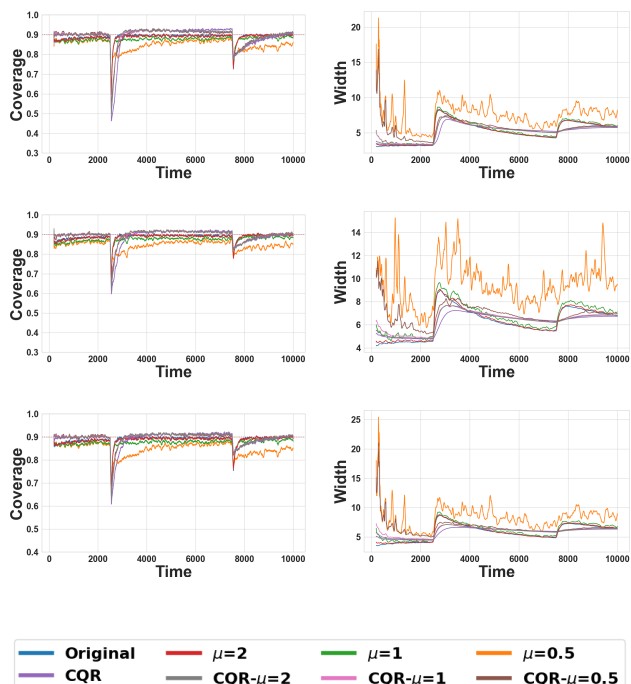

*Figure 2.* Simulation results for Setting 2 (Cases 1, 3, and 5, top to bottom) with $\mu$-GDP: The mean prediction interval coverage and width of our proposed algorithms and the Original algorithm are computed over 200 independent trials. For stability, the first 200 time points are excluded from the analysis, and the displayed curves are smoothed using a rolling average with a window size of 50 time points.

with data recorded from three inertial measurement units (IMUs) placed on the wrist, chest, and ankle, as well as heart rate (HR) monitoring data. For the analysis, we focus on the heart rate data from subject 102 as the target response variable. Heart rate is a highly individualized physiological signal that reflects an individual's health status and activity levels, making it particularly sensitive and critical in health-related contexts. Consequently, safeguarding the privacy of such data is essential for real-world applications.

We refer to our prediction strategy as adaptive lagged feature regression (ALFR), where a regression model is retrained at each time step using a recent history of lagged features to capture short-term temporal dependencies. In our implementation, we instantiate ALFR using a third-order autoregressive (AR(3)) model, following the setup of Angelopoulos et al. (2023). This design allows the model to adapt to non-stationary physiological signals.

In terms of coverage, we observe that the rolling coverage of our proposed algorithm improves as the privacy protection parameter $\mu$ decreases. Specifically, when $\mu = 1$, the rolling coverage exhibits the highest volatility, while the ODPCP variant without privacy noise achieves minimal volatility and stabilizes around the target confidence level

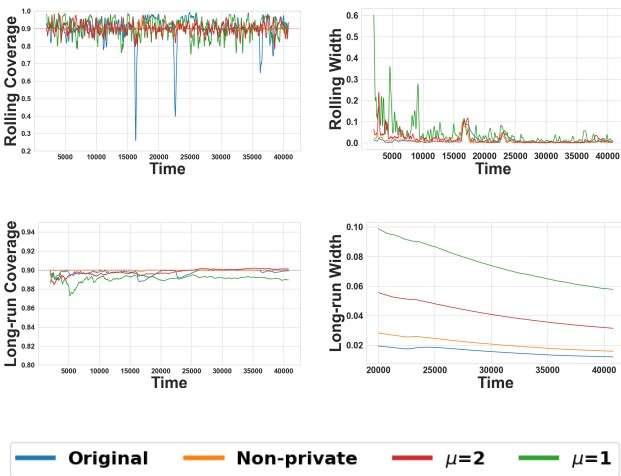

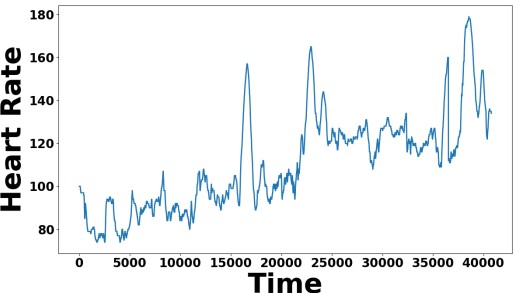

*Figure 4.* The trend of heart rate variation

*Figure 3.* **PAMAP2 dataset**: The upper panel displays the rolling coverage and rolling width, each averaged over a rolling window of 200 time points. The lower panel shows the long-run coverage, $\sum_{t=1}^{T} \mathbb{I}\{Y_t \in \hat{C}_t(X_t)\}/T$ and the long-run interval width, both averaged over 200 repetitions. For stability, the first 2000 time points are excluded from the analysis. The displayed curves are smoothed using a rolling average with a window size of 50 time points. **"Non-private"** refers to the version of our proposed ODPCP algorithm without privacy noise.

of $1 - \alpha = 0.9$. In contrast, the rolling coverage of the Original algorithm shows four distinct drops: the first two reduce coverage to approximately 0.3, while the subsequent two drops lower it to around 0.7. These fluctuations are closely associated with sudden changes in the data, as depicted in Figure 4, where sharp shifts at corresponding time points lead to the failure of the Original algorithm to adjust promptly, causing a sharp decline in coverage. However, the magnitude of the latter two drops is notably smaller than the first two, as the algorithm becomes more stable over time and better accommodates abrupt changes in the data. In comparison, our proposed method incorporates a truncation parameter $c$, which helps to stabilize coverage during data shifts. In terms of long-run coverage, all methods eventually converge to similar levels, indicating comparable steady-state performance.

Regarding interval width, the Original algorithm outperforms our proposed algorithm, with the rolling interval width exhibiting the highest stability, and the long-run interval width ultimately converging to the smallest value. On the PAMAP2 dataset, our algorithm follows a similar trend as observed in previous datasets: the perturbation introduced by the privacy mechanism significantly increases the volatility of the interval width, resulting in a larger long-run interval width. This indicates that, while the privacy protection mechanism enhances the stability of coverage, it comes at the cost of wider prediction intervals, thereby reducing the precision of the predictions. To provide a more intuitive

understanding of the resulting prediction intervals for heart rate, we include a visualization in Figure 21. Additional classification results on the PAMAP2 dataset are provided in Appendix C.4 to further demonstrate the versatility of our method.

## 7. Conclusion

In this paper, we introduced the online differentially private conformal prediction framework, which is designed to generate dynamic, model-free private prediction sets while ensuring robust privacy guarantees. Our approach operates as a one-pass algorithm, which eliminates the need for storing or re-accessing historical data, making it particularly well-suited for real-time, privacy-sensitive applications. Moreover, we extended our framework to incorporate conformal quantile regression, enabling the construction of adaptive prediction intervals in heteroscedastic settings. To assess the effectiveness of our proposed framework, we conducted extensive experiments, including both simulations and real-world case studies using the ELEC2 and PAMAP2 datasets. The results showed that our method provides reliable prediction set coverage, while effectively preserving privacy, even in the face of data distribution shifts and evolving patterns over time.

## Acknowledgements

We sincerely thank the anonymous reviewers and area chairs for their valuable feedback and constructive suggestions, which significantly improved the quality of this work. Qiangqiang Zhang and Xinwei Feng were supported by the National Key R&D Program of China (No. 2023YFA1008701) and the National Natural Science Foundation of China (Grant Nos. 12371148, 12326603, and 12431017). Ting Li's research was partially supported by the Shanghai Pujiang Programme (No. 24PJC030), CCF-DiDi GAIA Collaborative Research Funds and the Program for Innovative Research Team of Shanghai University of Finance and Economics. Xiaodong Yan was also supported by the National Key R&D Program of China (No. 2023YFA1008701) and the National Natural Science Foun-

dation of China (No. 12371292). Jinhan Xie's research was funded by the National Key R&D Program of China (No. 102022YFA1003701).

## Impact Statement

This paper presents a novel methodological framework that advances the field of machine learning, with a particular focus on privacy-preserving prediction techniques. By addressing key challenges in online private conformal prediction, this work contributes to the development of models that are not only more robust and interpretable but also prioritize privacy. The primary objective of this research is to make significant strides in machine learning in a principled, responsible manner, ensuring that privacy considerations are deeply integrated into predictive modeling. While the societal implications of this work are considerable, we believe these are inherently tied to the broader objective of promoting privacy and fairness in machine learning systems, and thus, do not need to be explicitly highlighted in this context.

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

# Appendix

## A. Algorithm of online differentially private conformal quantile

We extend our framework to incorporate Conformal Quantile Regression. The algorithm generates valid conformal quantile regression sets $\hat{C}_t(X_t)$ that simultaneously guarantee differential privacy and coverage probability.

---

**Algorithm 3** Online differentially private conformal quantile regression set

---

1: **Input:** Data $(X_t, Y_t), t = 1, 2, \ldots$; Prediction models $\hat{q}_{\alpha_{\mathrm{lo}}}(\cdot)$ and $\hat{q}_{\alpha_{\mathrm{hi}}}(\cdot)$; Privacy budget parameters $\epsilon_t > 0$; $\alpha \in (0,1)$; Constant $c > 0$; $S_1 > 0$
2: **for** $t = 2, 3, \ldots$ **do**
3:     Estimate the privatized quantile:
4:     $\hat{q}_t^{1-\alpha} \leftarrow$ Algorithm 2$(\epsilon_{t-1}, \alpha, S_{t-1}, c)$
5:     Predict and compute the non-conformity score:
6:     $S_t = \max\left\{\hat{q}_{\alpha_{\mathrm{lo}}}(X_t) - Y_t, Y_t - \hat{q}_{\alpha_{\mathrm{hi}}}(X_t)\right\}$
7: **end for**
8: **Output:** Private prediction sets:

$$\hat{C}_t(X_t) = \left[\hat{q}_{\alpha_{\mathrm{lo}}}(X_t) - \hat{q}_t^{1-\alpha}, \hat{q}_{\alpha_{\mathrm{hi}}}(X_t) + \hat{q}_t^{1-\alpha}\right].$$

---

## B. Proofs

*Proof of Theorem 4.4.* For the sake of simplicity, we demonstrate that Algorithm 1 satisfies $\max_{1 \leq j \leq t} \epsilon_j$-DP when using the Laplace mechanism. A similar argument holds for other DP mechanisms.

To make the argument more intuitive, we rewrite the update rule in Algorithm 2 as:

$$\hat{q}_{t+1}^{1-\alpha} = \left(\frac{t}{t+1}\lambda_t - \frac{1}{t+1}\hat{g}_t\right) \cdot \left(W_{t-1} - \left(\hat{g}_t\hat{q}_t^{1-\alpha}\right) \vee c\right),$$

where the privatized gradient is given by

$$\hat{g}_t = \partial\ell_{1-\alpha}(q, S_t)\big|_{q=\hat{q}_t^{1-\alpha}} + \mathcal{Z}_t, \quad \mathcal{Z}_t \sim \mathrm{Laplace}\left(0, \frac{\Delta_1(\partial\ell_{1-\alpha})}{\epsilon_t}\right).$$

For $t = 2$, the initial estimate $\hat{q}_1^{1-\alpha}$ is deterministic, and thus the output $\hat{q}_2^{1-\alpha}$ is $\epsilon_1$-DP by the standard Laplace mechanism and the post-processing property in Lemma 3.6. Since Algorithm 1 accesses the private quantile estimates only through Algorithm 2, its output constitutes a post-processing of differentially private computations. Hence, the prediction set $\hat{C}_2(X_2)$ is also $\epsilon_1$-DP.

For $t = 3$, since the update depends only on the disjoint data point $S_2$ and the previous output $\hat{q}_2^{1-\alpha}$, which is already private. By the parallel composition property in Lemma 3.6, the overall mechanism remains $\max\{\epsilon_1, \epsilon_2\}$-DP. Applying the same argument inductively yields that Algorithm 1 is $\max_{1 \leq j \leq t} \epsilon_j$-DP over $t$ update steps.

$\square$

*Proof of Theorem 4.5.* **1.** For simplicity, we prove this theorem in the context of $\mu$-GDP, noting that similar arguments can be extended to other mechanisms. In long-term iterations, the wealth process $W_t$, inspired by the Kelly criterion, exhibits a growth trend to asymptotic infinity even in adversarial environments. Consequently, there exists a time $T$ after which the practical constraint of $c$ naturally vanishes, leaving the optimization process governed by the adaptive mechanism(Orabona

& Pál, 2016; Cutkosky & Orabona, 2018). For theoretical simplicity, we omit $c$ in our analysis. Observe that $\mathcal{Z}_i$ follows a normal distribution, i.e., $\mathcal{Z}_i \sim N\left(0, \left(\frac{\Delta_2(h)}{\mu_i}\right)^2\right)$ for $i \geq 1$. Under the assumption that the nonconformity scores are bounded, specifically, $S_i \leq D$ for all $i = 1, 2, \ldots$, the following statements hold:

(a) Suppose that for some $i \geq 1$, the predicted radius $\hat{q}_i^{1-\alpha}$ exceeds the upper bound $D$, i.e., $\hat{q}_i^{1-\alpha} > D$. Given that $\hat{q}_i^{1-\alpha} = \lambda_i \cdot W_{i-1}$ and the wealth is nonnegative ($W_{i-1} \geq 0$), it follows that $\lambda_i > 0$. Moreover, the corresponding (sub)gradient is given by

$$g_i = \alpha - \mathbb{I}\{Y_i \notin \hat{C}_i(X_i)\} = \alpha - \mathbb{I}\{S_i > \hat{q}_i^{1-\alpha}\} = \alpha,$$

and the perturbed gradient is

$$\hat{g}_i = \alpha + \mathcal{Z}_i.$$

If $\hat{g}_i < 0$, then ,

$$W_i = W_{i-1}(1 - \lambda_i \hat{g}_i) > W_{i-1}.$$

It is known that,

$$\lambda_{i+1} = \frac{i}{i+1}\lambda_i - \frac{1}{i+1}\hat{g}_i.$$

At this stage, it is unclear whether $\lambda_i$ is greater or smaller than $\lambda_{i+1}$, making it impossible to determine the relationship between $\hat{q}_{i+1}^{1-\alpha}$ and $\hat{q}_i^{1-\alpha}$.

If $\hat{g}_i > 0$, then, conversely,

$$W_i = W_{i-1}(1 - \lambda_i \hat{g}_i) < W_{i-1}.$$

We thus obtain:

$$\lambda_{i+1} = \frac{i}{i+1}\lambda_i - \frac{1}{i+1}\hat{g}_i.$$

At this point, we can infer that $\lambda_{i+1} < \lambda_i$, which implies that $\hat{q}_{i+1}^{1-\alpha} > \hat{q}_i^{1-\alpha}$, resulting in:

$$\hat{q}_{i+1}^{1-\alpha} = \lambda_{i+1}W_i < \hat{q}_i^{1-\alpha}.$$

Consequently, since $\alpha > 0$, the probability that $\hat{g} > 0$ exceeds $\frac{1}{2}$. As a result, the overall trend is decreasing, which implies that the predicted radius eventually stabilizes and remains bounded by $D$.

(b) Suppose that for some $i \geq 1$, we have $\hat{q}_i^{1-\alpha} \geq 0$ but $\hat{q}_{i+1}^{1-\alpha} < 0$. This implies that there must exist some $k$ such that $\hat{q}_{i+k}^{1-\alpha} > 0$. Indeed, since $s_i \geq 0$, it follows that $\lambda_i \geq 0$, whereas $\hat{q}_{i+1}^{1-\alpha} < 0$ implies that $\lambda_{i+1} < 0$. Thus, we conclude:

$$0 > \lambda_{i+1} = \frac{i}{i+1}\lambda_i - \frac{1}{i+1}\hat{g}_i,$$

which implies that $\hat{g}_i > 0$. Given that $S_{i+1} \geq 0$, it follows that:

$$g_{i+1} = \alpha - \mathbb{I}\{S_{i+1} > \hat{q}_{i+1}^{1-\alpha}\} = \alpha - 1, \quad \hat{g}_{i+1} = \alpha - 1 + \mathcal{Z}_{i+1}.$$

Consequently, we derive:

$$\lambda_{i+2} = \frac{i+1}{i+2}\lambda_{i+1} - \frac{1}{i+2}\hat{g}_{i+1}$$

$$= \frac{i+1}{i+2} \cdot \frac{i}{i+1}\lambda_i - \frac{i+1}{i+2} \cdot \frac{1}{i+1}\hat{g}_i - \frac{1}{i+2}\hat{g}_{i+1}$$

$$= \frac{i}{i+2}\lambda_i - \frac{1}{i+2}(\hat{g}_i + \hat{g}_{i+1}).$$

$$= \frac{i}{i+2}\lambda_i - \frac{1}{i+2}(2\alpha - 1 + \mathcal{Z}_i + \mathcal{Z}_{i+1}) \quad \text{or} \quad = \frac{i}{i+2}\lambda_i - \frac{1}{i+2}(2\alpha - 2 + \mathcal{Z}_i + \mathcal{Z}_{i+1}).$$

In either case, we obtain,

$$g_i + g_{i+1} = 2\alpha - 1 < 0 \quad \text{and} \quad g_i + g_{i+1} = 2\alpha - 2 < 0,$$

assuming $\alpha < 0.5$ as a mild condition. Thus, $\lambda_i$ generally exhibits an increasing trend. We conclude that there exists some $k$ such that

$$\lambda_{i+k} > 0,$$

and hence,

$$\hat{q}_{i+k}^{1-\alpha} > 0.$$

**2.** Since for any $t \geq 1$, $W_t = 1 - \sum_{i=1}^{t} \hat{q}_i^{1-\alpha} \hat{g}_i \geq 0$, Hence,

$$\sum_{i=1}^{t} \hat{q}_i^{1-\alpha} \hat{g}_i \leq 1.$$

Therefore,

$$\sum_{i=1}^{t} \hat{g}_i \hat{q}_i^{1-\alpha} = \sum_{i=1}^{t} (g_i + \mathcal{Z}_i) \hat{q}_i^{1-\alpha} \cdot 1\{\hat{q}_i^{1-\alpha} > D\} + \sum_{i=1}^{t} (g_i + \mathcal{Z}_i) \hat{q}_i^{1-\alpha} \cdot 1\{\hat{q}_i^{1-\alpha} \in [0, D]\}$$

$$+ \sum_{i=1}^{t} (g_i + \mathcal{Z}_i) \hat{q}_i^{1-\alpha} \cdot 1\{\hat{q}_i^{1-\alpha} < 0\}$$

$$= \sum_{i=1}^{t} g_i \hat{q}_i^{1-\alpha} \cdot 1\{\hat{q}_i^{1-\alpha} > D\} + \sum_{i=1}^{t} g_i \hat{q}_i^{1-\alpha} \cdot 1\{\hat{q}_i^{1-\alpha} \in [0, D]\}$$

$$+ \sum_{i=1}^{t} g_i \hat{q}_i^{1-\alpha} \cdot 1\{\hat{q}_i^{1-\alpha} < 0\} + \sum_{i=1}^{t} \mathcal{Z}_i \hat{q}_i^{1-\alpha} \cdot 1\{\hat{q}_i^{1-\alpha} > D\}$$

$$+ \sum_{i=1}^{t} \mathcal{Z}_i \hat{q}_i^{1-\alpha} \cdot 1\{\hat{q}_i^{1-\alpha} \in [0, D]\} + \sum_{i=1}^{t} \mathcal{Z}_i \hat{q}_i^{1-\alpha} \cdot 1\{\hat{q}_i^{1-\alpha} < 0\}.$$

We know if $\hat{q}_i^{1-\alpha} > D$, then we have that $g_i = \alpha > 0$, and if $\hat{q}_i^{1-\alpha} < 0$, then $g_i = \alpha - 1 < 0$. Hence,

$$\sum_{i=1}^{t} \hat{g}_i \hat{q}_i^{1-\alpha} \geq \sum_{i=1}^{t} g_i \hat{q}_i^{1-\alpha} \cdot 1\{\hat{q}_i^{1-\alpha} \in [0, D]\} + \sum_{i=1}^{t} \mathcal{Z}_i \hat{q}_i^{1-\alpha} \cdot 1\{\hat{q}_i^{1-\alpha} > D\}$$

$$+ \sum_{i=1}^{t} \mathcal{Z}_i \hat{q}_i^{1-\alpha} \cdot 1\{\hat{q}_i^{1-\alpha} \in [0, D]\} + \sum_{i=1}^{t} \mathcal{Z}_i \hat{q}_i^{1-\alpha} \cdot 1\{\hat{q}_i^{1-\alpha} < 0\}$$

$$\geq -Dt + \sum_{i=1}^{t} \mathcal{Z}_i \hat{q}_i^{1-\alpha}.$$

We have shown that $-Dt + \sum_{i=1}^{t} \mathcal{Z}_i \hat{q}_i^{1-\alpha} \leq \sum_{i=1}^{t} \hat{q}_i^{1-\alpha} g_i \leq 1$, and hence,

$$\left| \sum_{i=1}^{t} \hat{q}_i^{1-\alpha} g_i \right| \leq \max \left\{ 1, Dt + \left| \sum_{i=1}^{t} \mathcal{Z}_i \hat{q}_i^{1-\alpha} \right| \right\} \leq Dt + \left| \sum_{i=1}^{t} \mathcal{Z}_i \hat{q}_i^{1-\alpha} \right| + 1. \tag{7}$$

Next, we bound the distance between the consecutive predicted radii. Note that:

$$\hat{q}_{t+1}^{1-\alpha} = -\frac{\sum_{i=1}^{t} \hat{g}_i}{t+1} \left( 1 - \sum_{i=1}^{t} \hat{g}_i \hat{q}_i^{1-\alpha} \right)$$

$$= -\frac{\sum_{i=1}^{t}\hat{g}_i}{t+1}\left(1-\sum_{i=1}^{t-1}\hat{g}_i\hat{q}_i^{1-\alpha}\right)+\hat{g}_t\hat{q}_t^{1-\alpha}\frac{\sum_{i=1}^{t}\hat{g}_i}{t+1}$$

$$= -\frac{\sum_{i=1}^{t-1}\hat{g}_i}{t+1}\left(1-\sum_{i=1}^{t-1}\hat{g}_i\hat{q}_i^{1-\alpha}\right)-\frac{\hat{g}_t}{t+1}\left(1-\sum_{i=1}^{t-1}\hat{g}_i\hat{q}_i^{1-\alpha}\right)+\frac{\hat{g}_t\hat{q}_t^{1-\alpha}\sum_{i=1}^{t}\hat{g}_i}{t+1}$$

$$= \frac{t}{t+1}\hat{q}_t^{1-\alpha}+\frac{1}{t+1}\left(-\hat{g}_t+\hat{g}_t\sum_{i=1}^{t-1}\hat{g}_i\hat{q}_i^{1-\alpha}+\hat{g}_t\hat{q}_t^{1-\alpha}\sum_{i=1}^{t}\hat{g}_i\right),$$

Thus,

$$\hat{q}_{t+1}^{1-\alpha}-\hat{q}_t^{1-\alpha}=\frac{1}{t+1}\left(-\hat{q}_t^{1-\alpha}-\hat{g}_t+\hat{g}_t\sum_{i=1}^{t-1}\hat{g}_i\hat{q}_i^{1-\alpha}+\hat{g}_t\hat{q}_t^{1-\alpha}\sum_{i=1}^{t}\hat{g}_i\right).$$

$$\hat{q}_{t+1}^{1-\alpha}-\hat{q}_t^{1-\alpha}=\frac{1}{t+1}\bigg(-\hat{q}_t^{1-\alpha}-g_t-\mathcal{Z}_t+\hat{g}_t\sum_{i=1}^{t-1}\hat{g}_i\hat{q}_i^{1-\alpha}$$

$$+g_t\hat{q}_t^{1-\alpha}\sum_{i=1}^{t}g_i+g_t\hat{q}_t^{1-\alpha}\sum_{i=1}^{t}\mathcal{Z}_i$$

$$+\mathcal{Z}_t\hat{q}_t^{1-\alpha}\sum_{i=1}^{t}\mathcal{Z}_i+\mathcal{Z}_t\hat{q}_t^{1-\alpha}\sum_{i=1}^{t}g_i\bigg).$$

It then follows that:

$$\left|\hat{q}_{t+1}^{1-\alpha}-\hat{q}_t^{1-\alpha}\right|\leq\frac{1}{t+1}\left|-\hat{q}_t^{1-\alpha}-g_t+\hat{g}_t\sum_{i=1}^{t-1}\hat{g}_i\hat{q}_i^{1-\alpha}+g_t\hat{q}_t^{1-\alpha}\sum_{i=1}^{t}g_i\right|$$

$$+\frac{1}{t+1}\left|-\mathcal{Z}_t+g_t\hat{q}_t^{1-\alpha}\sum_{i=1}^{t}\mathcal{Z}_i+\mathcal{Z}_t\hat{q}_t^{1-\alpha}\sum_{i=1}^{t}\mathcal{Z}_i+\mathcal{Z}_t\hat{q}_t^{1-\alpha}\sum_{i=1}^{t}g_i\right|.$$

Combining this with the fact that $s_1=0\in[0,D]$, the result from step 1, and (7), we obtain:

$$\left|\hat{q}_{t+1}^{1-\alpha}-\hat{q}_t^{1-\alpha}\right|\leq\frac{1}{t+1}\left(D+1+D(t-1)+\left|\sum_{i=1}^{t-1}\mathcal{Z}_i\hat{q}_i^{1-\alpha}\right|+1+Dt\right)$$

$$+\frac{1}{t+1}\left|\mathcal{Z}_t+g_t\hat{q}_t^{1-\alpha}\sum_{i=1}^{t}\mathcal{Z}_i+\mathcal{Z}_t\hat{q}_t^{1-\alpha}\sum_{i=1}^{t}\mathcal{Z}_i+\mathcal{Z}_t\hat{q}_t^{1-\alpha}\sum_{i=1}^{t}g_i\right|$$

$$\leq 2D+1+\frac{1}{t+1}\bigg(|\mathcal{Z}_t|+\left|g_t\hat{q}_t^{1-\alpha}\sum_{i=1}^{t}\mathcal{Z}_i\right|+\left|\mathcal{Z}_t\hat{q}_t^{1-\alpha}\sum_{i=1}^{t}\mathcal{Z}_i\right|$$

$$+\left|\mathcal{Z}_t\hat{q}_t^{1-\alpha}\sum_{i=1}^{t}g_i\right|+\left|\sum_{i=1}^{t-1}\mathcal{Z}_i\hat{q}_i^{1-\alpha}\right|\bigg).$$

we can directly apply the Cauchy-Schwarz inequality to absolute values and squares, namely:

$$\left|\sum_{i=1}^{t-1}\mathcal{Z}_i\hat{q}_i^{1-\alpha}\right|\leq\left(\sum_{i=1}^{t-1}\mathcal{Z}_i^2\right)^{\frac{1}{2}}\left(\sum_{i=1}^{t-1}(\hat{q}_t^{1-\alpha})^2\right)^{\frac{1}{2}}.$$

Thus,

$$\leq 2D + 1 + \frac{1}{t+1}\left(\left|\mathcal{Z}_t\right| + D\left|\sum_{i=1}^{t}\mathcal{Z}_i\right| + D\left|\mathcal{Z}_t\sum_{i=1}^{t}\mathcal{Z}_i\right| + D\left|\mathcal{Z}_t\sum_{i=1}^{t}g_i\right| + \left((t-1)^{\frac{1}{2}}D\right)\left(\sum_{i=1}^{t-1}\mathcal{Z}_i^2\right)^{\frac{1}{2}}\right).$$

$$|\hat{q}_t^{1-\alpha}| \leq 3D + 1 + \frac{1}{t+1}\left(\left|\mathcal{Z}_t\right| + D\left|\sum_{i=1}^{t}\mathcal{Z}_i\right| + D\left|\mathcal{Z}_t\sum_{i=1}^{t}\mathcal{Z}_i\right| + D\left|\mathcal{Z}_t\sum_{i=1}^{t}g_i\right| + \left((t-1)^{\frac{1}{2}}D\right)\left(\sum_{i=1}^{t-1}\mathcal{Z}_i^2\right)^{\frac{1}{2}}\right).$$

We analyze the asymptotic behavior of each term as $t \to \infty$.

i. Constant term: The term $3D + 1$ is independent of $t$ and remains constant.

ii. The term $\frac{1}{t+1}|\mathcal{Z}_t|$:

Since $\mathcal{Z}_t \sim N\left(0, \left(\frac{\Delta_2(h)}{\mu_t}\right)^2\right)$, we have

$$\mathbb{E}[|\mathcal{Z}_t|] = \frac{\Delta_2(h)}{\mu_t}\sqrt{\frac{2}{\pi}}.$$

Thus, the expectation of this term is

$$\frac{1}{t+1}\mathbb{E}[|\mathcal{Z}_t|] = O\left(\frac{1}{t+1}\right),$$

which vanishes as $t \to \infty$.

iii. The term $\frac{1}{t+1}D\left|\sum_{i=1}^{t}\mathcal{Z}_i\right|$:

Notice that the expectation satisfies

$$\mathbb{E}\left[\left|\sum_{i=1}^{t}\mathcal{Z}_i\right|\right] = O(\sqrt{t}).$$

Thus, we obtain

$$\frac{1}{t+1}DO(\sqrt{t}) = O\left(\frac{D\sqrt{t}}{t+1}\right).$$

As $t \to \infty$, this term vanishes at a rate of $O(D/\sqrt{t})$.

iv. The term $\frac{1}{t+1}D\left|\mathcal{Z}_t\sum_{i=1}^{t}\mathcal{Z}_i\right|$:

Using the Cauchy-Schwarz inequality:

$$\mathbb{E}\left[\left|\mathcal{Z}_t\sum_{i=1}^{t}\mathcal{Z}_i\right|\right] \leq \sqrt{\mathbb{E}[\mathcal{Z}_t^2]\cdot\mathbb{E}\left[\left(\sum_{i=1}^{t}\mathcal{Z}_i\right)^2\right]} = O\left(\sqrt{t}\right).$$

Thus, the term becomes

$$\frac{1}{t+1}DO\left(\sqrt{t}\right) = O\left(\frac{D}{\sqrt{t}}\right).$$

v. The term $\frac{1}{t+1}D\left|\mathcal{Z}_t\sum_{i=1}^{t}g_i\right|$:

Given $|g_i| \leq 1$ for all $i$, we have $\sum_{i=1}^{t}g_i = O(t)$. Then:

$$\mathbb{E}\left[\left|\mathcal{Z}_t\sum_{i=1}^{t}g_i\right|\right] \leq \mathbb{E}[|\mathcal{Z}_t|]\cdot O(t) = O(t).$$

Thus,

$$\frac{1}{t+1} DO(t) = O(D).$$

vi. The term $\frac{1}{t+1}(t-1)^{\frac{1}{2}} D \left( \sum_{i=1}^{t-1} \mathcal{Z}_i^2 \right)^{\frac{1}{2}}$:

By Jensen's inequality and $\sum_{i=1}^{t-1} \mathcal{Z}_i^2 \sim \left( \frac{\Delta_2(h)}{\mu_1} \right)^2 \chi_{t-1}^2$:

$$\mathbb{E}\left[ \left( \sum_{i=1}^{t-1} \mathcal{Z}_i^2 \right)^{\frac{1}{2}} \right] = O(\sqrt{t}),$$

yielding

$$\frac{1}{t+1} O(D\sqrt{t}) = O(D).$$

Combining all terms:

$$|\hat{q}_t^{1-\alpha}| \le 3D + 1 + O\left(\frac{D}{t+1}\right) + O\left(\frac{D}{\sqrt{t}}\right) + O\left(\frac{D}{\sqrt{t}}\right) + O(D) + O(D).$$

As $t \to \infty$, all vanishing terms disappear, leaving:

$$|\hat{q}_t^{1-\alpha}| = O(D).$$

Thus, $|\hat{q}_t^{1-\alpha}|$ asymptotically remains bounded by $O(D)$. Thus, as $t \to \infty$, the growth rate of $|\hat{q}_t^{1-\alpha}|$ stabilizes at the order of $O(D)$, indicating that $|\hat{q}_t^{1-\alpha}|$ remains bounded.

**3.** Assume that the long-term coverage guarantee fails, i.e., there exists $\kappa > 0$ such that:

$$\limsup_{T \to \infty} \left| \frac{1}{T} \sum_{t=1}^{T} \mathbb{I}\{Y_t \notin \hat{C}_t(X_t)\} - \alpha \right| \ge \kappa.$$

From the definition of $g_t = \alpha - \mathbb{I}\{Y_t \notin \hat{C}_t(X_t)\}$ and $\hat{g}_t = g_t + \mathcal{Z}_t$, we have:

$$\frac{1}{T} \sum_{t=1}^{T} \mathbb{I}\{Y_t \notin \hat{C}_t(X_t)\} = \alpha - \frac{1}{T} \sum_{t=1}^{T} g_t.$$

Substituting this into the coverage failure condition:

$$\limsup_{T \to \infty} \left| -\frac{1}{T} \sum_{t=1}^{T} g_t \right| \ge \kappa \implies \limsup_{T \to \infty} \frac{1}{T} \left| \sum_{t=1}^{T} g_t \right| \ge \kappa.$$

Since $\hat{g}_t = g_t + \mathcal{Z}_t$, we can write:

$$\frac{1}{T} \sum_{t=1}^{T} \hat{g}_t = \frac{1}{T} \sum_{t=1}^{T} g_t + \frac{1}{T} \sum_{t=1}^{T} \mathcal{Z}_t.$$

By the Law of Large Numbers for zero-mean noise ($\mathbb{E}[\mathcal{Z}_t] = 0$, $\text{Var}(\mathcal{Z}_t) < \infty$):

$$\frac{1}{T} \sum_{t=1}^{T} \mathcal{Z}_t \to 0 \quad \text{almost surely as } T \to \infty.$$

Thus, the coverage failure implies:

$$\limsup_{T \to \infty} \frac{1}{T} \left| \sum_{t=1}^{T} \hat{g}_t \right| \ge \kappa.$$

and then

$$\hat{q}_{T+1}^{1-\alpha} = \lambda_{T+1} W_T = \frac{1}{T+1} \left| \sum_{t=1}^{T} \hat{g}_t \right| \cdot W_T.$$

If $\frac{1}{T} \left| \sum_{t=1}^{T} \hat{g}_t \right| \geq \kappa$ infinitely often, then for such $T$:

$$|\hat{q}_{T+1}^{1-\alpha}| \geq \frac{T}{T+1} \kappa \cdot W_T.$$

From the wealth bound $W_T \geq \frac{1}{K\sqrt{T}} \exp\left( \frac{T}{4} \left( \frac{1}{T} \sum_{t=1}^{T} \hat{g}_t \right)^2 \right)$ (Orabona & Pál, 2016):

$$|\hat{q}_{T+1}^{1-\alpha}| \geq \frac{\kappa\sqrt{T}}{K(T+1)} \exp\left( \frac{T}{4}\kappa^2 \right) = O\left( \frac{1}{\sqrt{T}} \exp\left( \frac{T\kappa^2}{4} \right) \right).$$

This implies $|\hat{q}_{T+1}^{1-\alpha}| \to \infty$ as $T \to \infty$, contradicting the boundedness of $s_t$ derived earlier. We have completed the proof of this theorem. $\square$

## C. Additional experiments

### C.1. Simulation results for Setting 1

In this subsection, we provide simulation results for Setting 1.

#### C.1.1. SENSITIVITY ANALYSIS RESULTS FOR $c$

In Figure 5, we analyze the variations in coverage and interval length for different values of $c$ with $\mu$-GDP. These findings are in agreement with the results presented in the main text.

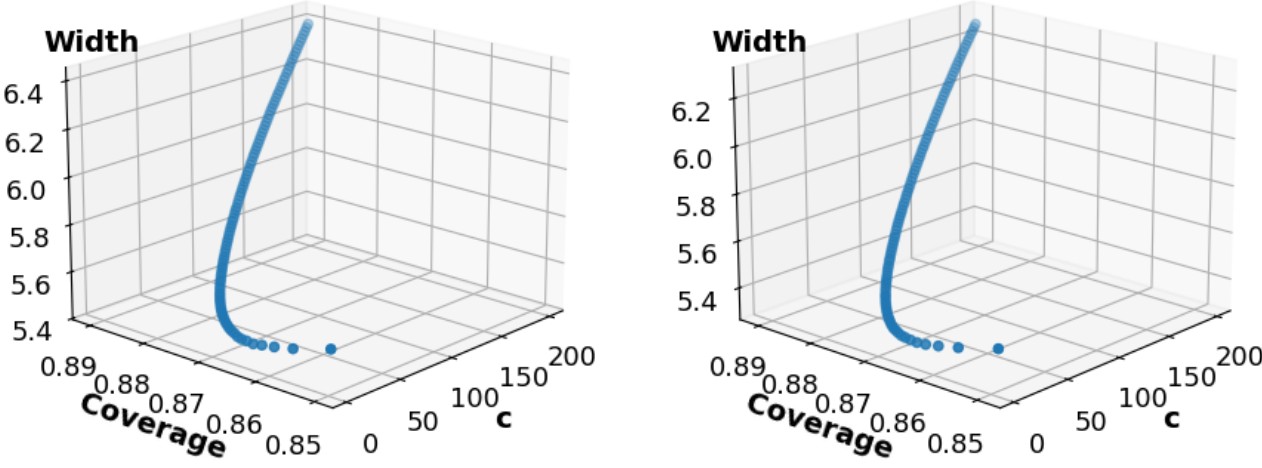

*Figure 5.* We simulate the coverage rate and interval length for different values of $c$ under the $\mu$-GDP setting. The value of $c$ ranges from 1 to 200 with an interval of 2. The results are based on the Setting 1 dataset. From left to right, the figures correspond to Case 1 and Case 2. For each case, we set the step length $t$ to 10000.

In Figure 6, we analyze the variations in coverage and coverage length for different values of $c$ with $\epsilon$-DP. Compared to the results obtained with the $\mu$-GDP, both exhibit nearly identical trends.

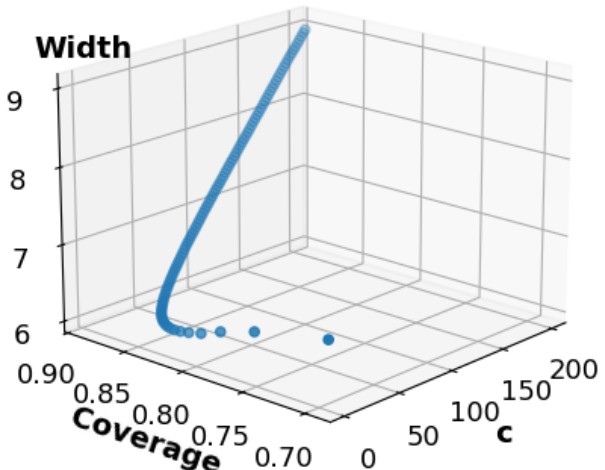 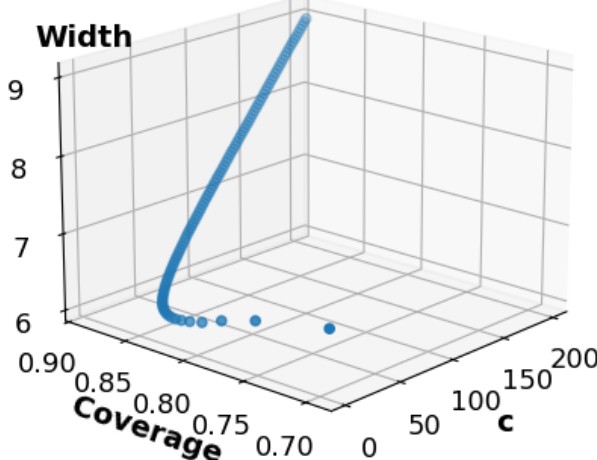

*Figure 6.* We simulate the coverage rate and interval length for different values of $c$ under the $\epsilon$-DP setting. The value of $c$ ranges from 1 to 200 with an interval of 2. The results are based on the Setting 1 dataset. From left to right, the figures correspond to Case 1 and Case 2. For each case, we set the step length $t$ to 10000.

*Table 3.* Long-run coverage $\sum_{t=1}^{T} \mathbb{I}\{Y_t \in \hat{C}_t(X_t)\}/T$ for the Setting 1 dataset, averaged over 200 independent trials with Gaussian noise, using both non-private and private base models. The first 50 data points are omitted to reduce the impact of early-stage instability.

| Case | Base Model | Original | $\mu = 2$ | $\mu = 1$ | $\mu = 0.5$ |
|------|-----------|----------|-----------|-----------|-------------|
| 1 | SGD | 0.890 | 0.887 | 0.876 | 0.866 |
|   | LDPSGD | 0.890 | 0.888 | 0.875 | 0.857 |
| 2 | SGD | 0.890 | 0.887 | 0.876 | 0.866 |
|   | LDPSGD | 0.890 | 0.888 | 0.874 | 0.856 |

### C.1.2. ADDITIONAL SIMULATION RESULTS FOR THE PRIVATELY TRAINED MODEL

We apply SGD and LDPSGD as the non-private and private base models, respectively, within the ODPCP framework, and present the corresponding results.

In Table 3, Table 4, and Figure 7, we report the experimental results of ODPCP under the $\mu$-GDP mechanism, using both non-private and private base models. The results show that the injected noise in LDPSGD leads to larger non-conformity scores compared to SGD, resulting in wider prediction intervals. Nevertheless, the coverage rate remains nearly unaffected.

In Table 5, Table 6, and Figure 8, we report the experimental results of ODPCP under the $\epsilon$-DP mechanism. The results are nearly identical to those under $\mu$-GDP. However, due to the larger magnitude of Laplace noise, the overall fluctuations in coverage and interval width are slightly greater than those observed with Gaussian noise.

### C.1.3. COMPARISON BETWEEN ODPCP AND ODPCQR

In Table 7, Table 8, and Figure 9, we present the experimental results of ODPCP and ODPCQR under the $\mu$-GDP noise mechanism. Regarding coverage, unlike Setting 2, neither algorithm experiences a sharp decline due to the absence of changepoints, instead maintaining coverage levels around $1 - \alpha = 0.9$ with minor fluctuations. In terms of interval length, both algorithms initially exhibit larger fluctuations but stabilize over time, demonstrating strong performance with stable and consistent prediction intervals in the later stages.

In Table 9, Table 10, and Figure 10, we present the experimental results of ODPCP and ODPCQR under the $\epsilon$-DP noise

*Table 4.* Long-run width for the Setting 1 dataset, averaged over 200 independent trials with Gaussian noise, using both non-private and private base models. The first 50 data points are omitted to reduce the impact of early-stage instability.

| Case | Base Model | Original | $\mu = 2$ | $\mu = 1$ | $\mu = 0.5$ |
|------|-----------|----------|-----------|-----------|-------------|
| 1 | SGD | 3.23 | 3.26 | 3.42 | 4.93 |
| | LDPSGD | 5.05 | 5.16 | 5.43 | 7.90 |
| 2 | SGD | 3.23 | 3.26 | 3.42 | 4.92 |
| | LDPSGD | 4.93 | 5.03 | 5.29 | 7.69 |

*Table 5.* Long-run coverage $\sum_{t=1}^{T} \mathbb{I}\{Y_t \in \hat{C}_t(X_t)\}/T$ for the Setting 1 dataset, averaged over 200 independent trials with Laplace noise, using both non-private and private base models. The first 100 data points are omitted to reduce the impact of early-stage instability.

| Case | Base Model | Original | $\epsilon = 2$ | $\epsilon = 1$ | $\epsilon = 0.5$ |
|------|-----------|----------|----------------|----------------|------------------|
| 1 | SGD | 0.890 | 0.884 | 0.869 | 0.866 |
| | LDPSGD | 0.890 | 0.884 | 0.859 | 0.852 |
| 2 | SGD | 0.890 | 0.885 | 0.869 | 0.866 |
| | LDPSGD | 0.890 | 0.884 | 0.859 | 0.853 |

mechanism. Since Laplace noise has a heavier tail distribution compared to Gaussian noise, the resulting perturbations are more substantial, leading to slightly more pronounced fluctuations in coverage. Similarly, in terms of interval length, the fluctuations are more noticeable under Laplace noise, yet the model still exhibits a relatively high level of convergence.

### C.2. Additional simulation results for Setting 2

Here, we present additional simulation results for Setting 2.

#### C.2.1. SENSITIVITY ANALYSIS RESULTS FOR $c$

In Figure 11 we analyze the variations in coverage and interval length for different values of $c$. These findings are in agreement with the results presented in the main text.

#### C.2.2. COMPARISON BETWEEN ODPCP AND DPCP

DPCP is an offline approach for constructing prediction sets under differential privacy, requiring access to the full calibration dataset. However, due to its static model, it is not suitable for streaming data or distribution shifts, such as changepoints. As shown in Table 11, while DPCP achieves slightly higher coverage, our method (ODPCP) consistently yields narrower prediction intervals, particularly under strong privacy constraints. This highlights the favorable privacy–efficiency trade-off offered by ODPCP in the online setting.

#### C.2.3. ADDITIONAL SIMULATION RESULTS FOR THE PRIVATELY TRAINED MODEL

We apply SGD and LDPSGD as the non-private and private *base models*, respectively, within the ODPCP framework, and present the corresponding results.

Table 12, Table 13, and Figure 12 present the simulation results for Setting 2 with $\mu$-GDP, reporting the coverage rate and interval length for the ODPCP algorithm under six different scenarios. These results are based on differentially private and non-private base models.

The experimental findings indicate that the injected noise in LDPSGD leads to larger non-conformity scores compared to SGD, resulting in wider prediction intervals. Nevertheless, the coverage rate remains nearly unaffected.

Table 14,Table 15 and Figure 13 present the simulation results for Setting 2 with $\epsilon - DP$, showing the coverage rate and

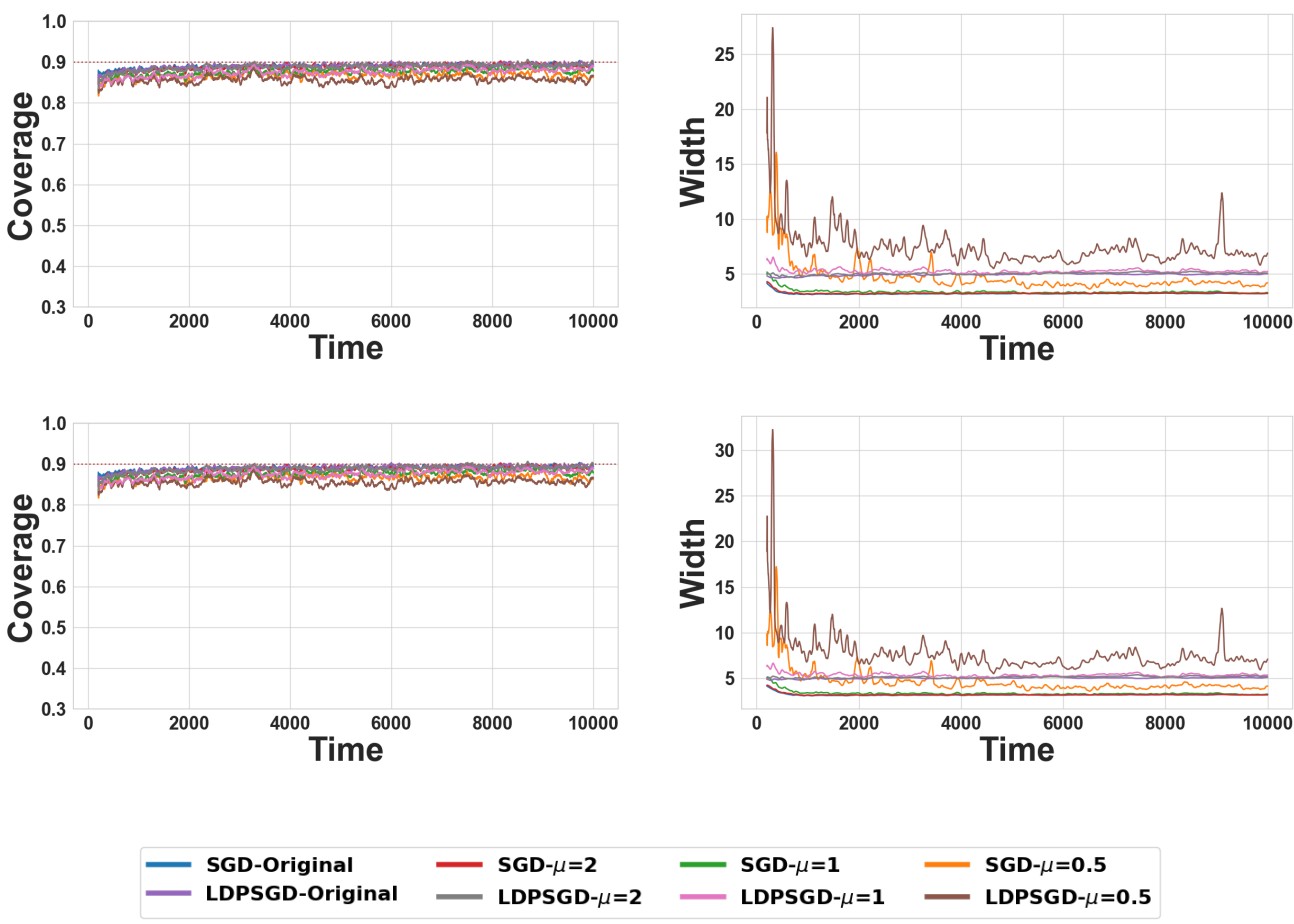

*Figure 7.* Simulation results for Setting 1 with $\mu$-GDP. The mean prediction interval coverage and width of our proposed algorithms, along with the Original version, are evaluated over 200 independent trials across two cases, using both non-private and private base models. To ensure stability, the first 200 time points are excluded from the analysis, and the displayed curves are smoothed using a rolling average with a window size of 50.

interval length for the ODPCP algorithm under six different scenarios, using both non-private and private base models. The results are nearly identical to those under $\mu$-GDP. However, due to the larger magnitude of Laplace noise, the overall fluctuations in coverage and interval width are slightly greater than those observed with Gaussian noise.

### C.2.4. COMPARISON BETWEEN ODPCP AND ODPCQR

Figure 14 presents the complementary results for Cases 2, 4, and 6, which, together with Figure 2 (Cases 1, 3, and 5), constitute the complete results of Setting 2 under the $\mu$-GDP mechanism. Table 16 ,Table 17 and Figure 15 present the simulation results for Setting 2 with $\epsilon - DP$. Compared to the results with Gaussian noise, the Laplace noise results exhibit greater fluctuations in coverage rate and interval length. However, in vertical comparison, the results are nearly identical to those with Gaussian noise, and we will not elaborate further; please refer to the main text.

### C.2.5. ROBUSTNESS TO INITIALIZATION

We evaluate the sensitivity of our method to the choice of initialization parameters $(\lambda_1, W_0)$. As shown in Figure 16, the performance remains virtually unchanged across a wide range of initialization settings. This confirms that the algorithm is highly robust and does not rely on careful tuning of initial values.

*Table 6.* Long-run width for the Setting 1 dataset, averaged over 200 independent trials with Laplace noise, using both non-private and private base models. The first 100 data points are omitted to reduce the impact of early-stage instability.

| Case | Base Model | Original | $\epsilon = 2$ | $\epsilon = 1$ | $\epsilon = 0.5$ |
|------|-----------|----------|--------|--------|----------|
| 1 | SGD | 3.22 | 3.30 | 3.80 | 7.65 |
| | LDPSGD | 5.05 | 5.22 | 5.93 | 12.43 |
| 2 | SGD | 3.22 | 3.30 | 3.80 | 7.71 |
| | LDPSGD | 4.93 | 5.09 | 5.80 | 12.30 |

*Table 7.* Long-Run Coverage $\sum_{t=1}^{T} \mathbb{I}\{Y_t \in \hat{C}_t(X_t)\}/T$, and the standard deviation (scaled by a factor of 1,000), are reported for the Setting 1 dataset with Gaussian noise, averaged over 200 independent trials .To minimize the impact of early-stage noise and the initial instability of the algorithm, the first 100 data points are excluded from the analysis.

| Case | Method | Original | $\mu = 2$ | $\mu = 1$ | $\mu = 0.5$ |
|------|--------|----------|-----------|-----------|-------------|
| 1 | CP | 0.890(0.26) | 0.888 (5.1) | 0.877(10) | 0.867 (19) |
| | CQR | 0.900 (1.7) | 0.900(4.7) | 0.900(9.3) | 0.899(19) |
| 2 | CP | 0.890 (0.27) | 0.888(5.2) | 0.877 (10) | 0.867(19) |
| | CQR | 0.900 (2.0) | 0.900(4.6) | 0.899(9.2) | 0.899(14) |

C.2.6. EFFECT OF DYNAMIC PRIVACY BUDGET ALLOCATION

As shown in Table 18, the random allocation of per-step privacy budgets ($\mu_t \sim \mathcal{U}(0.5, 2.0)$) yields nearly identical long-run coverage and interval width to the fixed high-budget case ($\mu_t = 2.0$) across all synthetic settings. This indicates that the maximum value of $\mu_t$ effectively dominates the resulting predictive uncertainty. In parallel, Figure 17 illustrates the asymptotic stability of ODPCP under increasing time horizons ($T = 10^4$ to $10^5$) across three configurations. The random-$\mu$ trajectory closely follows the fixed-high-$\mu$ baseline, further confirming that long-run behavior is determined by $\max_t \mu_t$, in alignment with Theorem 4.4.

## C.3. ElEC2 results

We evaluate the performance of our proposed algorithm using the ELEC2 dataset (Harries, 1999), which records electricity demand and pricing in the state of New South Wales, Australia, from May 1996 to December 1998. For this analysis, we focus exclusively on NSWdemand, which represents electricity demand in New South Wales. This variable is a critical indicator of market dynamics, directly reflecting fluctuations in electricity demand over time. Electricity demand data is inherently sensitive, as it captures both market trends and individual user behavior. In particular, large-scale historical demand data can be exploited to infer user behavior or predict future market trends, raising potential privacy concerns related to both commercial interests and personal privacy.

To mitigate these privacy risks, we apply our proposed method to this dataset. This ensures that individual or market-level information cannot be inferred, either directly or indirectly, during the data analysis process. In particular, we consider an autoregressive model of order 3 (AR(3)) to model the electricity demand data. A rolling window of size 200 is employed to compute dynamic coverage rates and prediction interval widths at each time step. The level of privacy protection is controlled by the parameter $\mu$, and we evaluate the model under two privacy budgets $\mu = 1, 2$.The results are reported in Figure 18.

In terms of coverage, our proposed algorithm, compared to its Original algorithm, exhibits greater volatility in both long-run and rolling coverage as the level of privacy protection increases. Specifically, under different privacy protection parameters ($\mu = 1, 2$), the rolling coverage fluctuates around the target confidence level of $1 - \alpha = 0.9$, with more pronounced variations than the Original algorithm. This effect is especially noticeable for $\mu = 1$, where the introduction of stronger

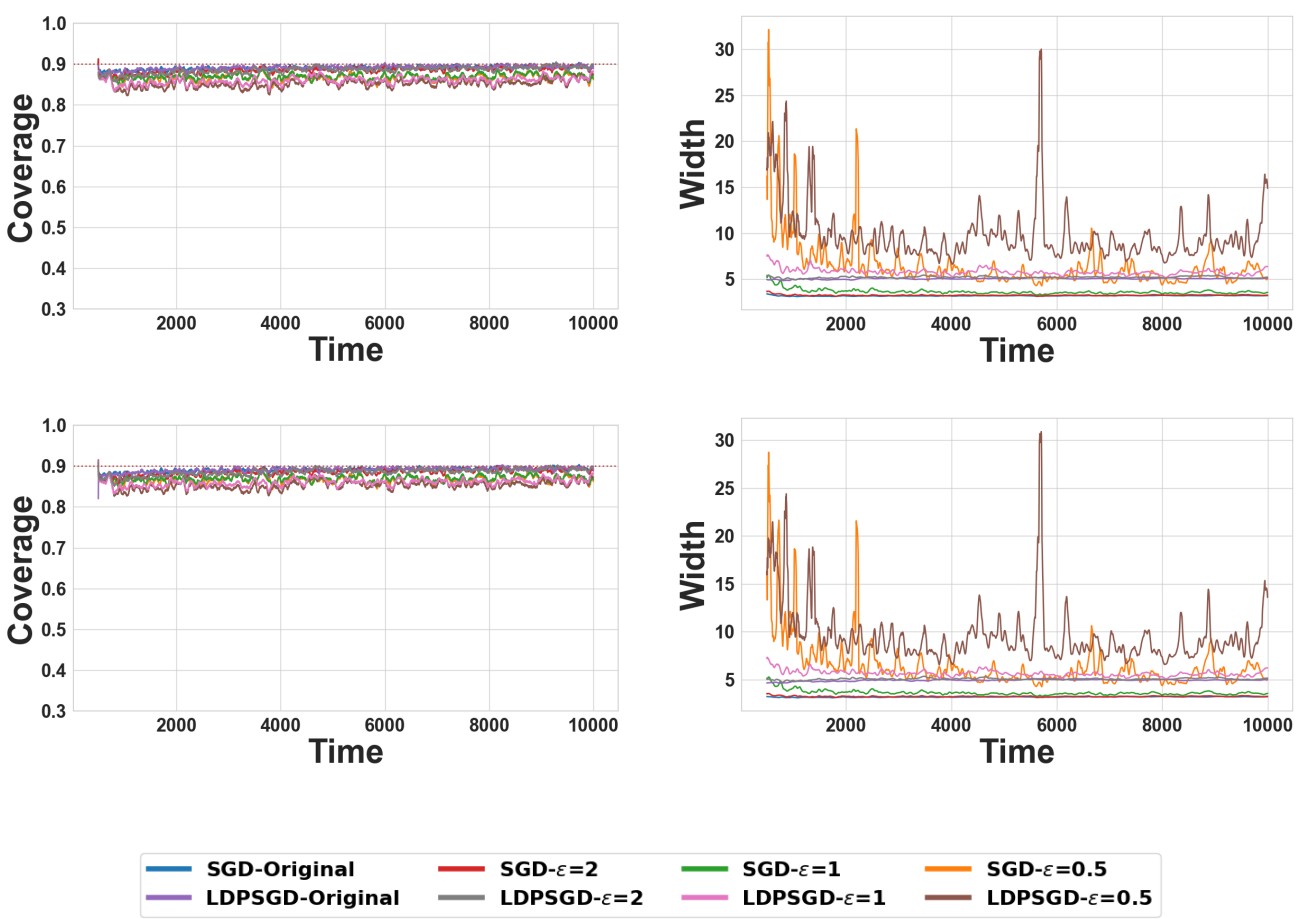

*Figure 8.* Simulation results for Setting 1 with $\epsilon$-DP. The mean prediction interval coverage and width of our proposed algorithms, along with the Original algorithm, are evaluated over 200 independent trials across two cases, using both non-private and private base models. To ensure stability, the first 500 time points are excluded from the analysis, and the displayed curves are smoothed using a rolling average with a window size of 50.

privacy protection leads to increased random perturbations, resulting in more significant fluctuations in coverage. Hence, higher levels of privacy protection tend to destabilize the coverage. For long-run coverage, as the time steps increase, the coverage volatility of the Original algorithm gradually decreases and stabilizes around the target confidence level. In contrast, our proposed algorithm with privacy protection shows greater volatility in the early stages, particularly for $\mu = 1$. However, this volatility decreases over time, with the coverage eventually converging to a stable value, albeit at a slower rate than Original algorithm.

With respect to the interval width, the rolling interval width of our proposed algorithm exhibits more significant fluctuations in the early stages but stabilizes as time progresses, eventually oscillating around the interval width of the Original algorithm. In terms of long-run interval width, all methods converge to stable values. However, the introduced noise due to privacy protection alters the final stable width. Specifically, when the privacy protection parameter $\mu$ is smaller (e.g., $\mu = 1$), the long-run interval width is noticeably larger than under other settings, with its Original version achieving the smallest interval width. This suggests that the noise perturbation introduced by the privacy protection mechanism results in a more conservative prediction interval, reducing the precision of the interval and thus its effectiveness in capturing the true values.

Figure 19 presents the results for the ElEC2 dataset under Laplace noise. The width and coverage exhibit larger fluctuations compared to Gaussian noise, with the results oscillating around those of the Original version. As the time step $t$ increases, stability improves and fluctuations decrease, a trend that is more pronounced in the rolling width.

*Table 8.* Long-Run Width, and the Standard Deviation (Scaled by 10), are reported for the Setting 1 dataset with Gaussian noise, averaged over 200 independent trials. To minimize the impact of early-stage noise and the initial instability of the algorithm, the first 100 data points are excluded from the analysis.

| Case | Method | Original | $\mu = 2$ | $\mu = 1$ | $\mu = 0.5$ |
|------|--------|----------|-----------|-----------|-------------|
| 1 | CP | 3.22(0.27) | 3.25(0.57) | 3.42(1.7) | 4.87 (15) |
|   | CQR | 3.31(0.34) | 3.33(0.55) | 3.39(1.2) | 4.00 (12) |
| 2 | CP | 3.22(0.27) | 3.26(0.56) | 3.42 (1.7) | 5.18 (42) |
|   | CQR | 3.31(0.32) | 3.33 (0.52) | 3.39(1.0) | 4.00 (23) |

*Table 9.* Long-Run Coverage $\sum_{t=1}^{T} \mathbb{I}\{Y_t \in \hat{C}_t(X_t)\}/T$ and the standard deviation (scaled by a factor of 1,000) are reported for the Setting 1 dataset with Laplace noise. To minimize the impact of early-stage noise and the initial instability of the algorithm, the first 100 data points are excluded from the analysis.

| Case | Method | Original | $\epsilon = 2$ | $\epsilon = 1$ | $\epsilon = 0.5$ |
|------|--------|----------|----------------|----------------|------------------|
| 1 | CP | 0.890 (0.27) | 0.884 (6.9) | 0.868 (13) | 0.865(25) |
|   | CQR | 0.900 (1.7) | 0.900(6.3) | 0.800 (12) | 0.899(25) |
| 2 | CP | 0.890 (0.27) | 0.884 (7.2) | 0.868 (14) | 0.866(27) |
|   | CQR | 0.900 (2.00) | 0.900 (6.7) | 0.900(13) | 0.899(27) |

## C.4. Additional PAMAP2 results

We evaluate ODPCP on activity classification using the PAMAP2 dataset, categorizing activities into three classes (resting, light, vigorous) based on heart rate and sensor data. An XGBoost model provides predictions, with ODPCP generating private prediction sets at each time step t. Figure 20 shows ODPCP's broad applicability to discrete prediction problems, maintaining strong empirical coverage and adaptive behavior under privacy constraints. Figure 22 presents the results for the PAMAP2 Physical Activity Monitoring Dataset under Laplace noise. The width and coverage exhibit larger fluctuations compared to Gaussian noise. However, as the algorithm progresses, the fluctuations decrease, demonstrating good convergence. Despite these fluctuations, the overall performance remains outstanding.

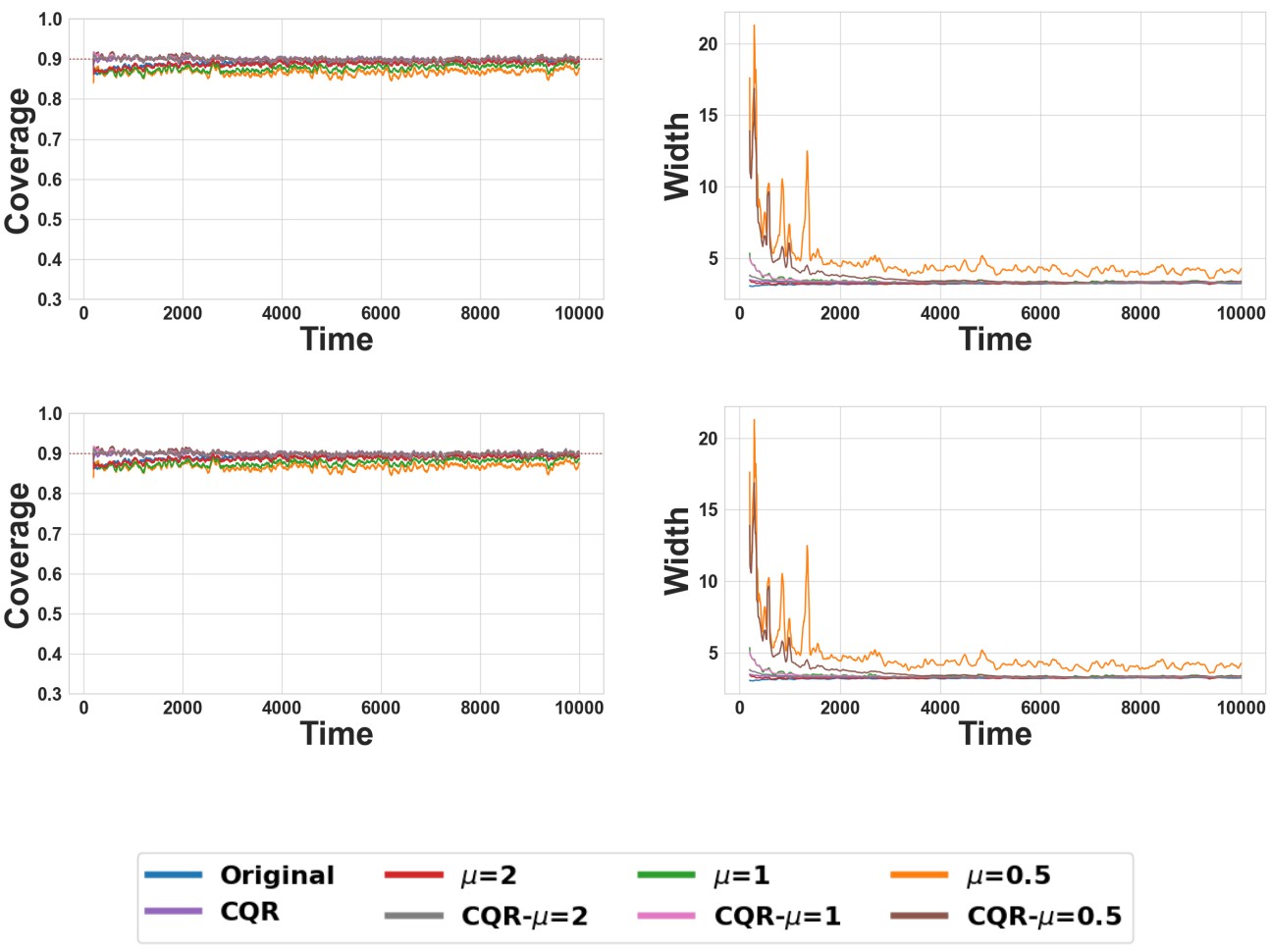

*Figure 9.* Simulation results for Setting 1 with $\mu$-GDP: The mean prediction interval coverage and width of our proposed algorithms, along with those of the Original algorithm, are evaluated over 200 independent trials across two cases. For stability, the first 200 time points are excluded from the analysis, and the displayed curves are smoothed using a rolling average with a window size of 50 time points.

*Table 10.* Long-Run Width and the Standard Deviation (Scaled by 10) are reported for the Setting 1 dataset with Laplace noise. To minimize the impact of early-stage noise and the initial instability of the algorithm, the first 100 data points are excluded from the analysis.

| Case | Method | Original | $\epsilon = 2$ | $\epsilon = 1$ | $\epsilon = 0.5$ |
|------|--------|----------|----------------|----------------|------------------|
| 1 | CP | 3.22(0.28) | 3.30(0.86) | 3.79(4.8) | 7.6 (48) |
| | CQR | 3.31(0.34) | 3.34(0.75) | 3.51(3.7) | 5.21(34) |
| 2 | CP | 3.22(0.27) | 3.30(0.84) | 3.82(4.6) | 9.4(149) |
| | CQR | 3.31(0.32) | 3.35(0.75) | 3.51(2.9) | 6.36(111) |

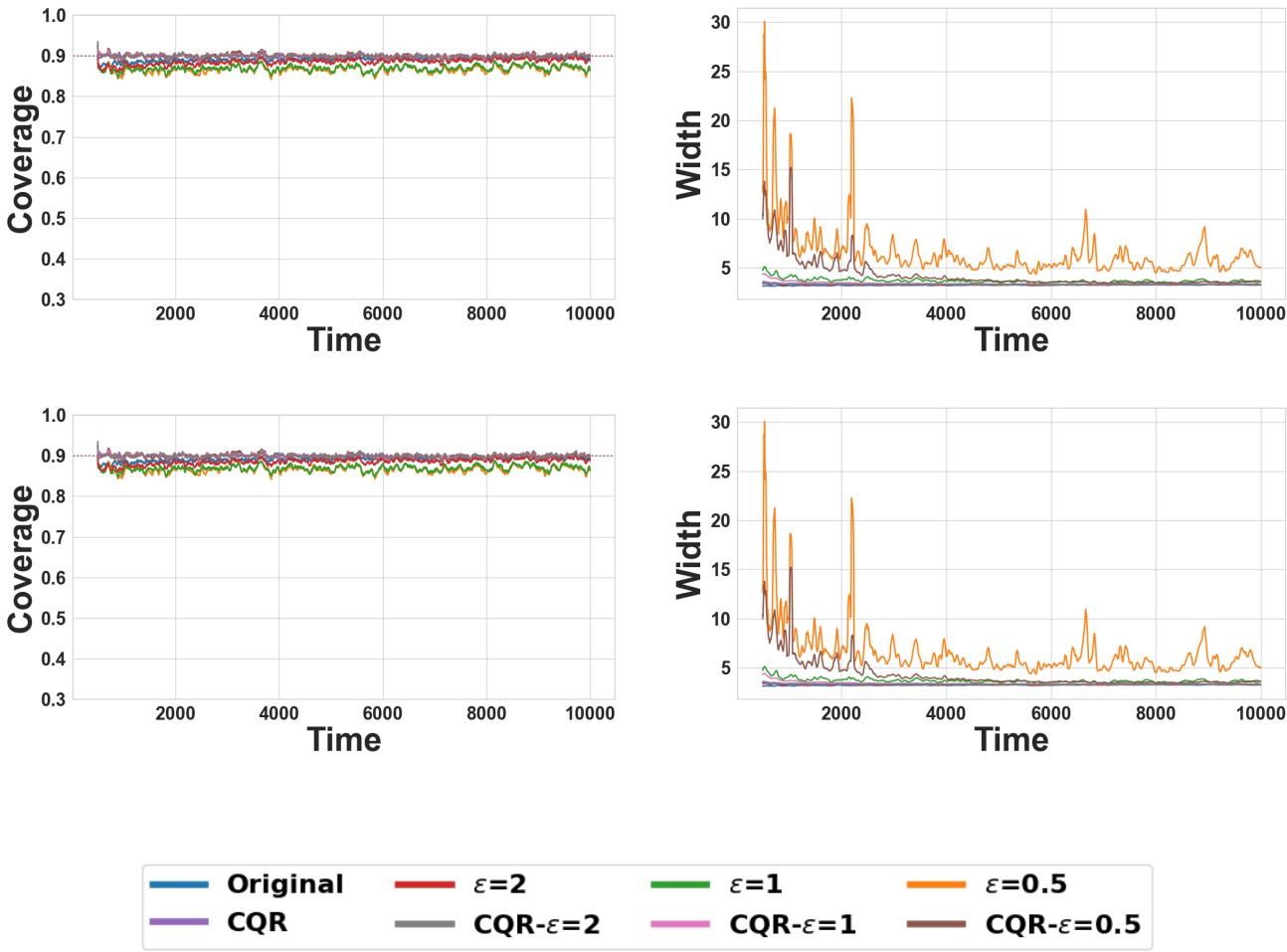

*Figure 10.* Simulation results for Setting 1 with $\epsilon$-DP: The mean prediction interval coverage and width for our proposed algorithms, along with the Original algorithm, are evaluated over 200 independent trials across two cases. To ensure stability, the first 500 time points are excluded from the analysis, and the displayed curves are smoothed using a rolling average with a window size of 50 time points.

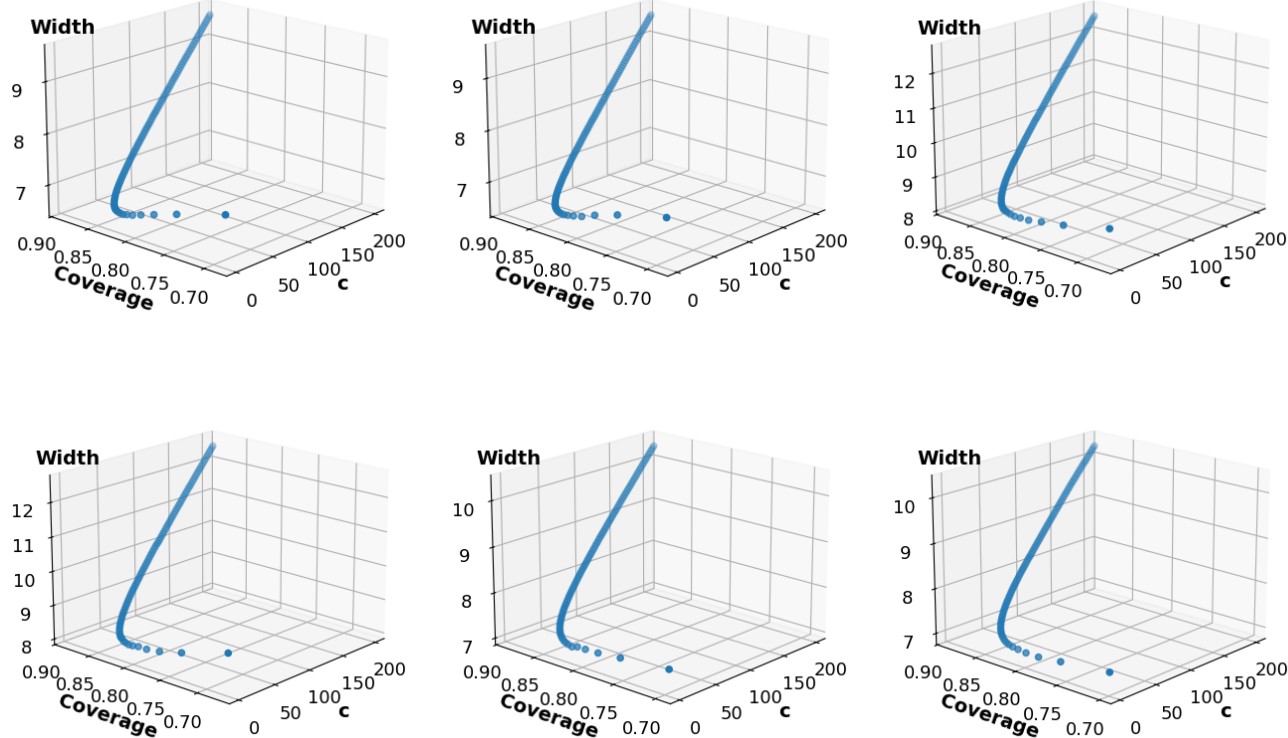

*Figure 11.* We simulate the coverage rate and interval length for different values of $c$ under the $\epsilon$-DP setting. The value of $c$ ranges from 1 to 200 with an interval of 2. Here, we present the results for the Setting 2 dataset. From left to right, the figures correspond to the six data settings, Case 1 to Case 6. The step length is set to $t = 10000$ for all cases.

*Table 11.* Comparison between ODPCP (our method) and DPCP ([Angelopoulos et al., 2022](#)) under Setting 2, across six simulation cases. Both methods target a nominal coverage level of 90% ($\alpha = 0.1$). The total sample size is $N = 10,000$.

| Case | Method | $\epsilon = 2$ | | $\epsilon = 1$ | |
|---|---|---|---|---|---|
| | | Coverage | Width | Coverage | Width |
| 1 | ODPCP | 0.885 | 3.65 | 0.868 | 4.20 |
| | DPCP | 0.905 | 5.58 | 0.911 | 5.68 |
| 2 | ODPCP | 0.885 | 3.70 | 0.868 | 4.26 |
| | DPCP | 0.906 | 7.16 | 0.911 | 7.28 |
| 3 | ODPCP | 0.885 | 5.04 | 0.863 | 5.81 |
| | DPCP | 0.906 | 6.66 | 0.911 | 6.80 |
| 4 | ODPCP | 0.885 | 5.11 | 0.864 | 5.89 |
| | DPCP | 0.906 | 8.03 | 0.912 | 8.24 |
| 5 | ODPCP | 0.884 | 4.50 | 0.869 | 4.96 |
| | DPCP | 0.907 | 6.31 | 0.912 | 6.49 |
| 6 | ODPCP | 0.884 | 4.59 | 0.869 | 5.04 |
| | DPCP | 0.903 | 7.69 | 0.908 | 7.89 |

*Table 12.* Long-run coverage $\sum_{t=1}^{T} \mathbb{I}\{Y_t \in \hat{C}_t(X_t)\}/T$ for the Setting 2 dataset, averaged over 200 independent trials with $\mu$-GDP, using both non-private and private base models. The first 50 data points are omitted to reduce the impact of early-stage instability.

| Case | Base Model | Original | $\mu = 2$ | $\mu = 1$ | $\mu = 0.5$ |
|------|-----------|----------|-----------|-----------|-------------|
| 1 | SGD | 0.890 | 0.888 | 0.876 | 0.865 |
|   | LDPSGD | 0.890 | 0.888 | 0.874 | 0.854 |
| 2 | SGD | 0.890 | 0.888 | 0.876 | 0.865 |
|   | LDPSGD | 0.890 | 0.888 | 0.875 | 0.855 |
| 3 | SGD | 0.890 | 0.888 | 0.875 | 0.860 |
|   | LDPSGD | 0.890 | 0.887 | 0.871 | 0.846 |
| 4 | SGD | 0.890 | 0.888 | 0.875 | 0.860 |
|   | LDPSGD | 0.890 | 0.887 | 0.871 | 0.847 |
| 5 | SGD | 0.890 | 0.887 | 0.876 | 0.864 |
|   | LDPSGD | 0.890 | 0.888 | 0.874 | 0.854 |
| 6 | SGD | 0.890 | 0.887 | 0.876 | 0.865 |
|   | LDPSGD | 0.890 | 0.888 | 0.874 | 0.855 |

*Table 13.* Long-run width for the Setting 2 dataset, averaged over 200 independent trials with $\mu$-GDP, using both non-private and private base models. The first 50 data points are omitted to reduce the impact of early-stage instability.

| Case | Base Model | Original | $\mu = 2$ | $\mu = 1$ | $\mu = 0.5$ |
|------|-----------|----------|-----------|-----------|-------------|
| 1 | SGD | 3.57 | 3.61 | 3.78 | 5.43 |
|   | LDPSGD | 5.42 | 5.54 | 5.83 | 8.46 |
| 2 | SGD | 3.62 | 3.66 | 3.85 | 5.50 |
|   | LDPSGD | 5.35 | 5.45 | 5.74 | 8.28 |
| 3 | SGD | 4.89 | 4.97 | 5.26 | 7.67 |
|   | LDPSGD | 6.65 | 6.72 | 7.09 | 10.2 |
| 4 | SGD | 4.95 | 5.04 | 5.33 | 7.75 |
|   | LDPSGD | 6.58 | 6.65 | 7.01 | 10.0 |
| 5 | SGD | 4.42 | 4.46 | 4.58 | 6.24 |
|   | LDPSGD | 5.97 | 6.11 | 6.44 | 9.15 |
| 6 | SGD | 4.51 | 4.55 | 4.66 | 6.33 |
|   | LDPSGD | 5.90 | 6.04 | 6.35 | 8.98 |

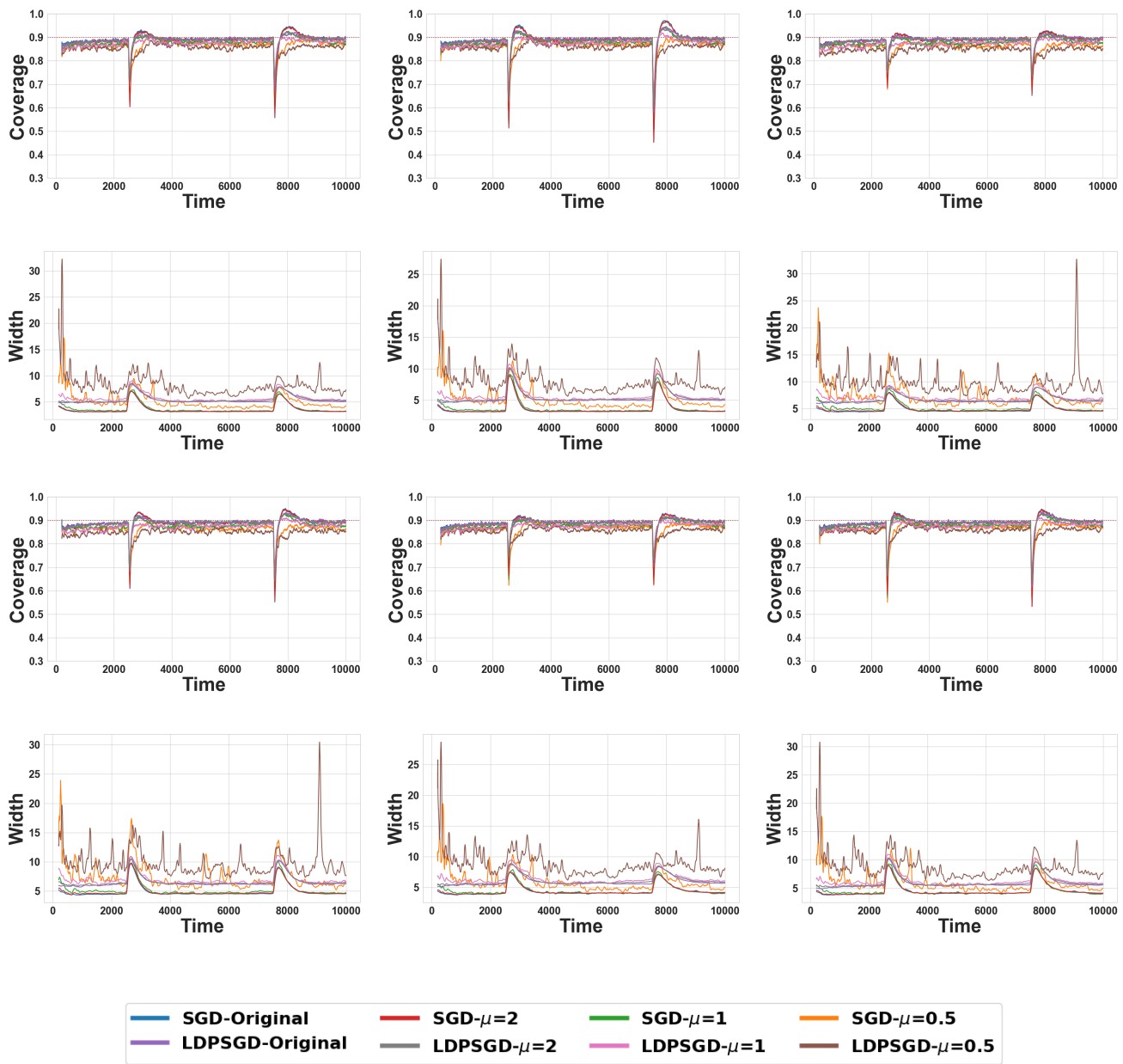

*Figure 12.* Simulation results for Setting 2 with $\mu$-GDP: The mean prediction interval coverage and width of our proposed algorithms, along with the Original algorithm, are evaluated over 200 independent trials across six cases (from left to right), using both non-private and private base models. To ensure stability, the first 200 time points are excluded from the analysis, and the displayed curves are smoothed using a rolling average with a window size of 50.

*Table 14.* Long-Run Coverage $\sum_{t=1}^{T} \mathbb{I}\{Y_t \in \hat{C}_t(X_t)\}/T$ for the Setting 2 dataset, averaged over 200 independent trials with Laplace noise, using both non-private and private base models. To reduce the excessive disturbance caused by early noise and algorithm instability, the first 100 data points are omitted, without affecting the convergence results.

| Case | Base Model | Original | $\epsilon = 2$ | $\epsilon = 1$ | $\epsilon = 0.5$ |
|------|-----------|----------|---------------|---------------|-----------------|
| 1 | SGD | 0.891 | 0.885 | 0.868 | 0.864 |
|   | LDPSGD | 0.890 | 0.884 | 0.858 | 0.849 |
| 2 | SGD | 0.891 | 0.885 | 0.868 | 0.864 |
|   | LDPSGD | 0.890 | 0.884 | 0.859 | 0.851 |
| 3 | SGD | 0.890 | 0.885 | 0.863 | 0.859 |
|   | LDPSGD | 0.890 | 0.883 | 0.851 | 0.840 |
| 4 | SGD | 0.890 | 0.885 | 0.864 | 0.860 |
|   | LDPSGD | 0.890 | 0.883 | 0.851 | 0.841 |
| 5 | SGD | 0.890 | 0.884 | 0.869 | 0.864 |
|   | LDPSGD | 0.890 | 0.883 | 0.858 | 0.848 |
| 6 | SGD | 0.890 | 0.884 | 0.869 | 0.863 |
|   | LDPSGD | 0.890 | 0.884 | 0.859 | 0.850 |

*Table 15.* Long-Run Width for the Setting 2 dataset, averaged over 200 independent trials with Laplace noise, using both non-private and private base models. To reduce the excessive disturbance caused by early noise and algorithm instability, the first 100 data points are omitted, without affecting the convergence results.

| Case | Base Model | Original | $\epsilon = 2$ | $\epsilon = 1$ | $\epsilon = 0.5$ |
|------|-----------|----------|---------------|---------------|-----------------|
| 1 | SGD | 3.56 | 3.65 | 4.20 | 8.30 |
|   | LDPSGD | 5.42 | 5.59 | 6.34 | 13.1 |
| 2 | SGD | 3.61 | 3.70 | 4.26 | 8.43 |
|   | LDPSGD | 5.35 | 5.52 | 6.27 | 13.0 |
| 3 | SGD | 4.88 | 5.04 | 5.81 | 11.7 |
|   | LDPSGD | 6.66 | 6.76 | 7.80 | 17.3 |
| 4 | SGD | 4.95 | 5.11 | 5.89 | 12.0 |
|   | LDPSGD | 6.59 | 6.70 | 7.73 | 17.0 |
| 5 | SGD | 4.42 | 4.50 | 4.96 | 9.41 |
|   | LDPSGD | 5.97 | 6.18 | 6.93 | 14.0 |
| 6 | SGD | 4.51 | 4.59 | 5.04 | 9.55 |
|   | LDPSGD | 5.90 | 6.11 | 6.84 | 14.0 |

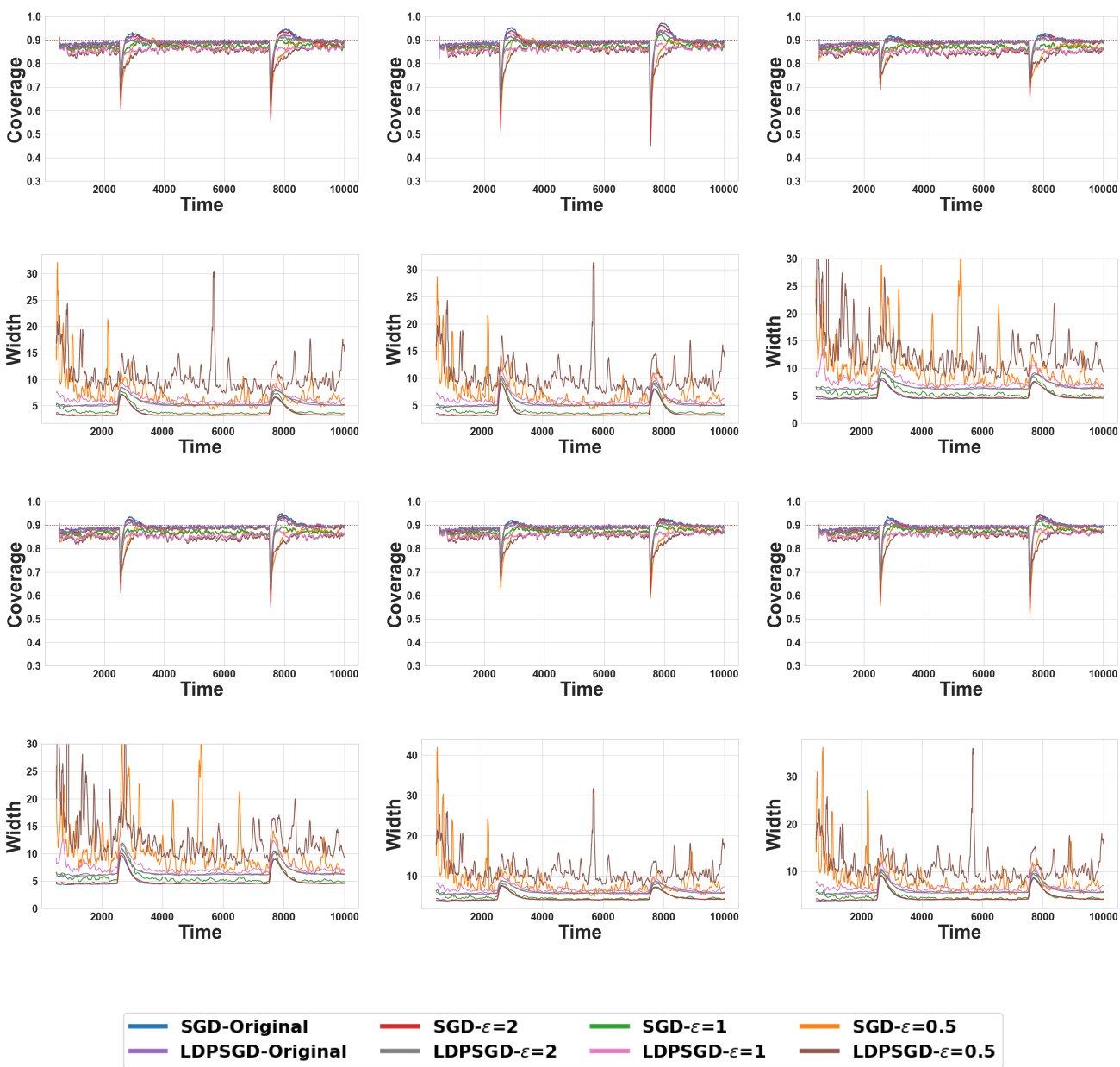

*Figure 13.* Simulation results for Setting 2 with $\epsilon$-DP: The mean prediction interval coverage and width of our proposed algorithms, along with the Original algorithm, are evaluated over 200 independent trials across six cases (from left to right), using both non-private and private base models. To ensure stability, the first 500 time points are excluded from the analysis, and the displayed curves are smoothed using a rolling average with a window size of 50 time points.

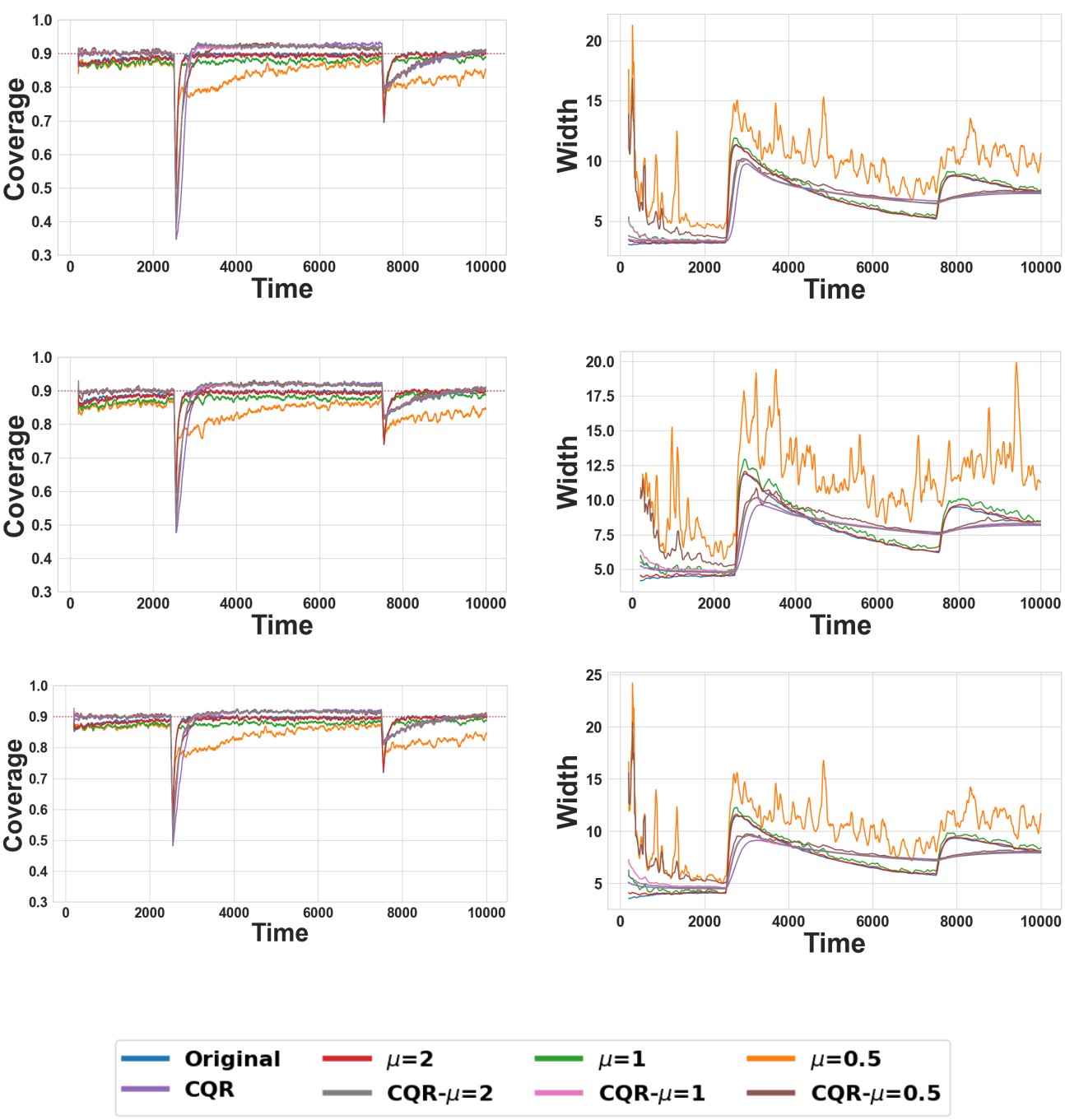

*Figure 14.* Simulation results for Setting 2 (Cases 2, 4, and 6, top to bottom) with $\mu$-GDP: The mean prediction interval coverage and width of our proposed algorithms and the Original algorithm are computed over 200 independent trials. For stability, the first 200 time points are excluded from the analysis, and the displayed curves are smoothed using a rolling average with a window size of 50 time points.

*Table 16.* Long-Run Coverage $\sum_{t=1}^{T} \mathbb{I}\{Y_t \in \hat{C}_t(X_t)\}/T$ and the standard deviation (scaled by a factor of 1,000) are reported for the Setting 2 dataset with Laplace noise. To minimize the impact of early-stage noise and the initial instability of the algorithm, the first 50 data points are excluded from the analysis.

| Case | Method | Original | $\epsilon = 2$ | $\epsilon = 1$ | $\epsilon = 0.5$ |
|------|--------|----------|----------------|----------------|------------------|
| 1 | CP | 0.889 (0.29) | 0.883 (6.9) | 0.857 (13) | 0.845 (25) |
|   | CQR | 0.890 (1.2) | 0.895 (6.2) | 0.895 (13) | 0.894 (25) |
| 2 | CP | 0.888 (0.28) | 0.882 (7.0) | 0.851 (13) | 0.836 (25) |
|   | CQR | 0.891 (1.4) | 0.895 (6.4) | 0.895 (12) | 0.894 (25) |
| 3 | CP | 0.889 (0.31) | 0.884 (8.7) | 0.853 (16) | 0.845 (24) |
|   | CQR | 0.892 (1.6) | 0.897 (6.9) | 0.897 (14) | 0.898 (28) |
| 4 | CP | 0.889 (0.26) | 0.883 (8.7) | 0.848 (16) | 0.833 (29) |
|   | CQR | 0.893 (1.5) | 0.896 (5.6) | 0.896 (11) | 0.896 (22) |
| 5 | CP | 0.889 (0.36) | 0.883 (7.0) | 0.856 (13) | 0.844 (25) |
|   | CQR | 0.892 (1.7) | 0.896 (5.6) | 0.896 (12) | 0.895 (23) |
| 6 | CP | 0.889 (0.35) | 0.882 (7.0) | 0.851 (12) | 0.834 (24) |
|   | CQR | 0.891 (1.8) | 0.895 (5.8) | 0.896 (10) | 0.897 (20) |

*Table 17.* Long-Run Width and the Standard Deviation (Scaled by 10) are reported for the Setting 2 dataset with Laplace noise. To minimize the impact of early-stage noise and the initial instability of the algorithm, the first 100 data points are excluded from the analysis.

| Case | Method | Original | $\epsilon = 2$ | $\epsilon = 1$ | $\epsilon = 0.5$ |
|------|--------|----------|----------------|----------------|------------------|
| 1 | CP | 5.19 (0.37) | 5.31 (1.2) | 6.13 (8.9) | 12.4 (83) |
|   | CQR | 5.11 (0.46) | 5.21 (1.0) | 5.48 (6.3) | 7.48 (38) |
| 2 | CP | 6.49 (0.65) | 6.63 (1.8) | 7.60 (10) | 15.3 (104) |
|   | CQR | 6.39 (0.68) | 6.49 (1.5) | 6.77 (6.5) | 8.91 (39) |
| 3 | CP | 6.21 (0.62) | 6.41 (2.5) | 7.41 (11) | 14.8 (84) |
|   | CQR | 6.22 (0.72) | 6.34 (1.8) | 6.58 (5.0) | 8.61 (37) |
| 4 | CP | 7.46 (0.80) | 7.67 (2.8) | 8.84 (13) | 21.8 (379) |
|   | CQR | 7.40 (0.79) | 7.52 (1.6) | 7.69 (3.6) | 11.8 (184) |
| 5 | CP | 5.79 (0.97) | 5.96 (2.5) | 6.85 (10) | 13.5 (87) |
|   | CQR | 5.85 (1.1) | 5.97 (2.0) | 6.27 (6.9) | 8.46 (41) |
| 6 | CP | 7.05 (1.1) | 7.23 (2.9) | 8.24 (12) | 16.5 (119) |
|   | CQR | 7.04 (1.1) | 7.19 (2.3) | 7.48 (6.8) | 9.44 (34) |

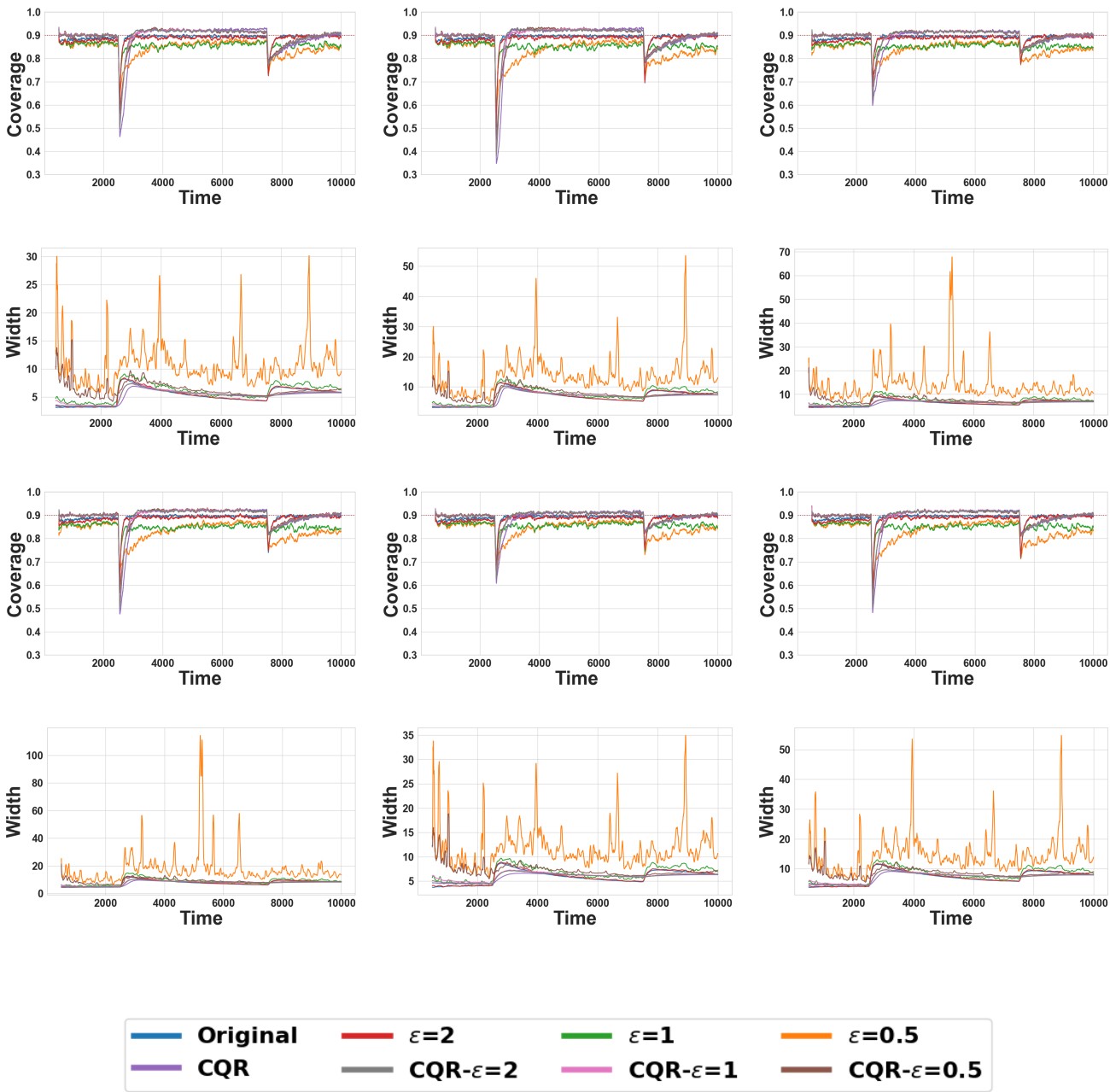

*Figure 15.* Simulation results for Setting 2 with $\epsilon$-DP: The mean prediction interval coverage and width for our proposed algorithms, along with the Original algorithm, are computed over 200 independent trials across six cases (from left to right), using both non-private and private base models. To ensure stability, the first 500 time points are excluded from the analysis, and the displayed curves are smoothed using a rolling average with a window size of 50 time points.

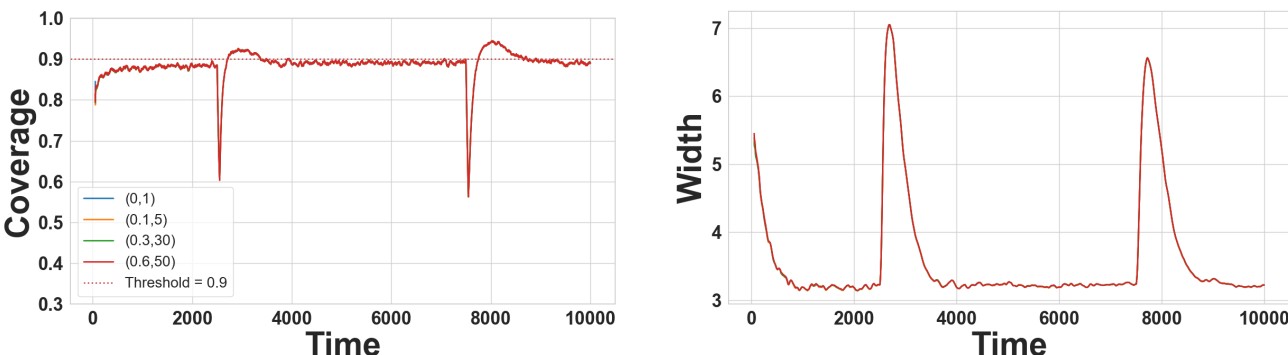

*Figure 16.* Effect of initialization $(\lambda_1, W_0)$ on coverage and interval width over a time horizon of $T = 10,000$. Curves under different initialization settings almost completely overlap, indicating high robustness to initialization.

*Table 18.* Long-run coverage and width under three privacy noise settings: fixed $\mu = 0.5$, fixed $\mu = 2.0$, and randomly sampled $\mu \sim \text{Uniform}(0.5, 2.0)$. Results are averaged over 200 runs with $T = 20,000$.

| Case | Metric | $\mu = 2$ | $\mu \in [0.5, 2]$ | $\mu = 0.5$ |
|------|--------|-----------|--------------------|-------------|
| 1 | Coverage | 0.895 | 0.887 | 0.872 |
|   | Width | 3.32 | 3.39 | 4.35 |
| 2 | Coverage | 0.894 | 0.886 | 0.870 |
|   | Width | 3.33 | 3.40 | 4.35 |
| 3 | Coverage | 0.894 | 0.885 | 0.860 |
|   | Width | 4.69 | 4.82 | 6.01 |
| 4 | Coverage | 0.894 | 0.886 | 0.861 |
|   | Width | 4.72 | 4.84 | 6.13 |
| 5 | Coverage | 0.894 | 0.886 | 0.870 |
|   | Width | 4.29 | 4.31 | 5.02 |
| 6 | Coverage | 0.894 | 0.884 | 0.869 |
|   | Width | 4.30 | 4.32 | 4.92 |

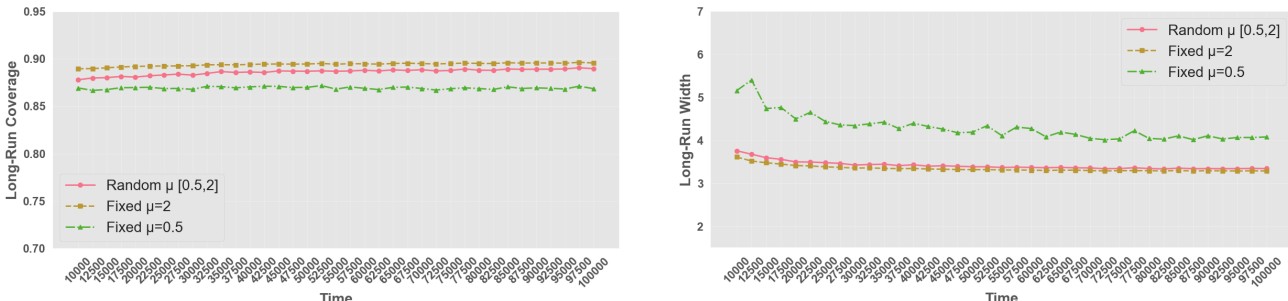

*Figure 17.* Asymptotic behavior of coverage and width under three privacy settings: fixed $\mu = 0.5$, fixed $\mu = 2.0$, and randomly sampled $\mu \sim \mathcal{U}(0.5, 2.0)$. Results show that the long-run performance under random $\mu$ closely follows that of fixed $\mu = 2.0$, indicating that the maximum $\mu_t$ largely determines the privacy–utility trade-off.

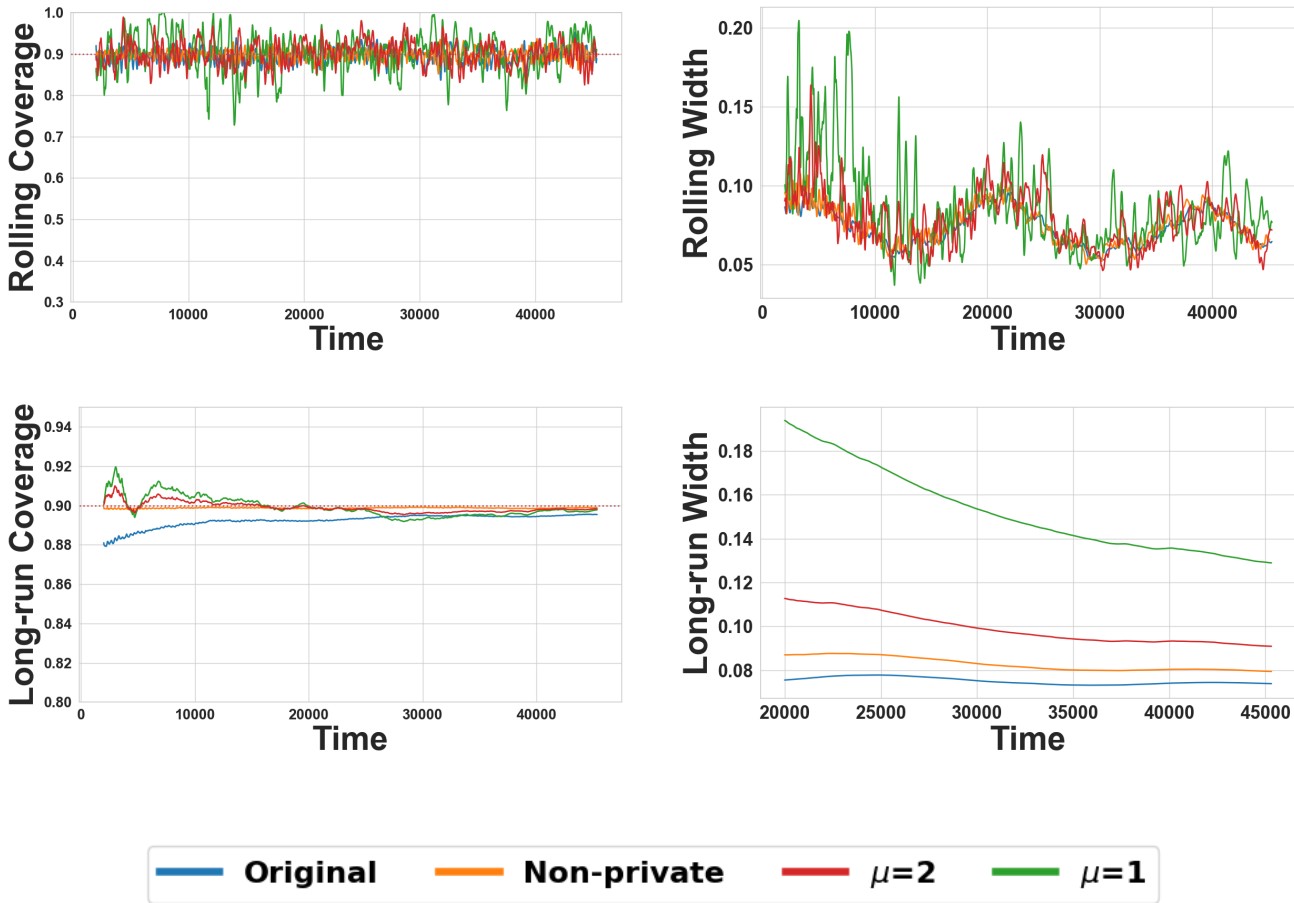

*Figure 18.* **ElEC2 dataset with $\mu$-GDP**: The upper panel illustrates the rolling coverage and rolling interval width, each averaged over a rolling window of 200 time points. For lower panel, we present the long-run coverage, defined as $\sum_{t=1}^{T} \mathbb{I}\{Y_t \in \hat{C}_t(X_t)\}/T$, and the long-run interval width, both averaged over 200 repetitions. For stability, the first 2000 time points are excluded from the analysis. For improved clarity and to mitigate fluctuations, the displayed curves are smoothed using a rolling average with a window size of 50 time points.

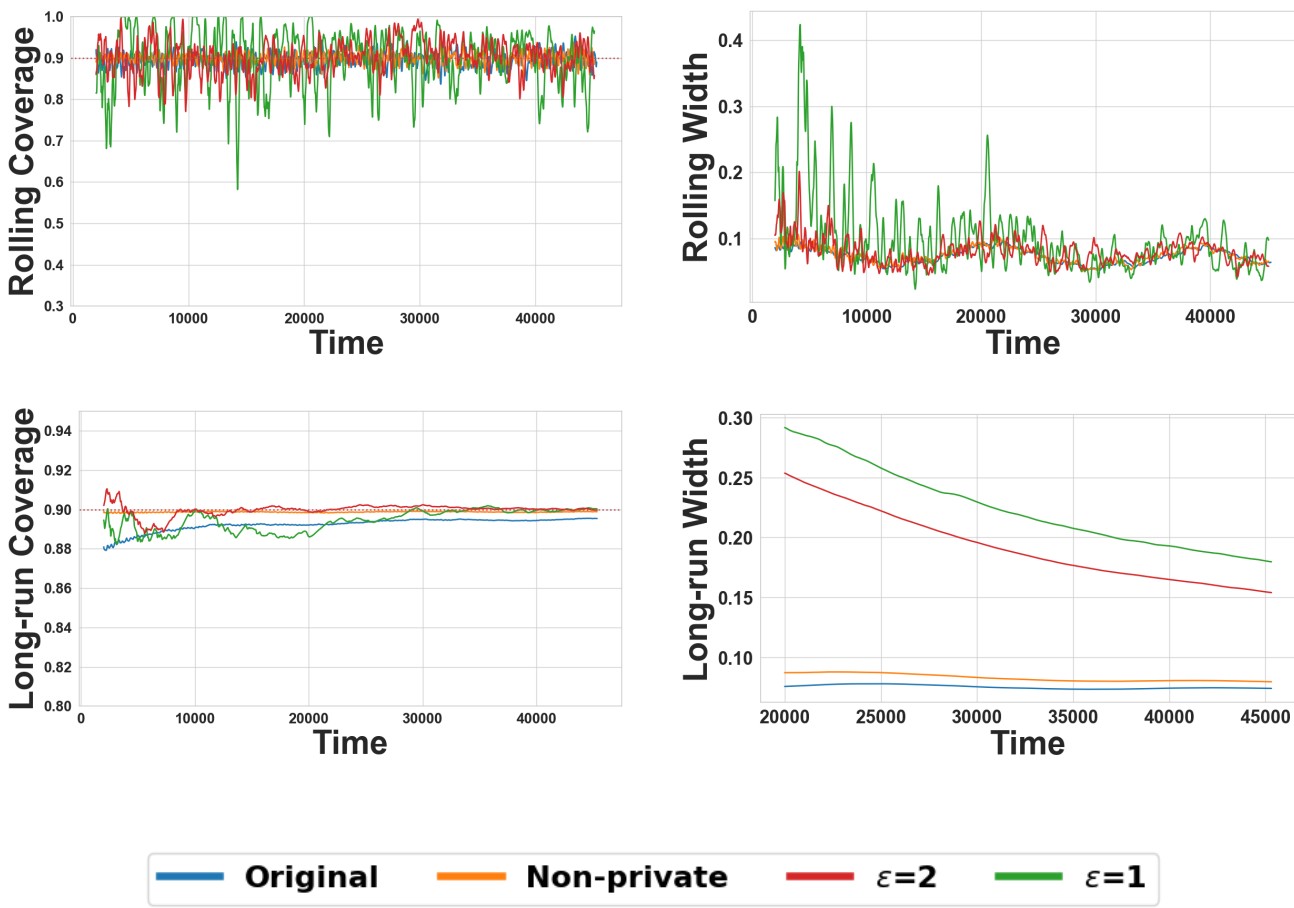

*Figure 19.* **ElEC2 Results with $\epsilon$-DP**. The first row shows the rolling coverage and rolling width, averaged over a rolling window of 200 time points. The second row displays the long-run coverage $\sum_{t=1}^{T} \mathbb{I}\{Y_t \in \hat{C}_t(X_t)\}/T$ and the long-run interval width, averaged over 200 repetitions. For stability, the first 2000 time points are excluded from the analysis. The displayed curves are smoothed by taking a rolling average with a window of 50 time points.

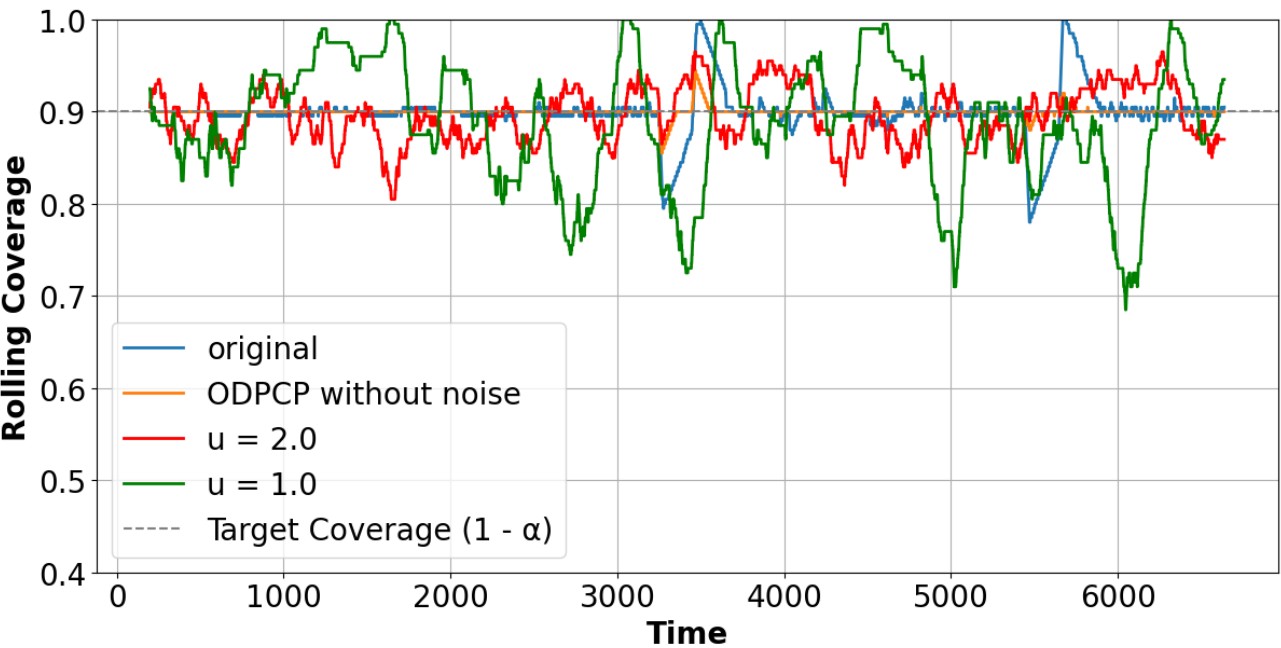

*Figure 20.* Multi-class classification experiment on subject 103 from the PAMAP2 dataset with a rolling window size of 200.

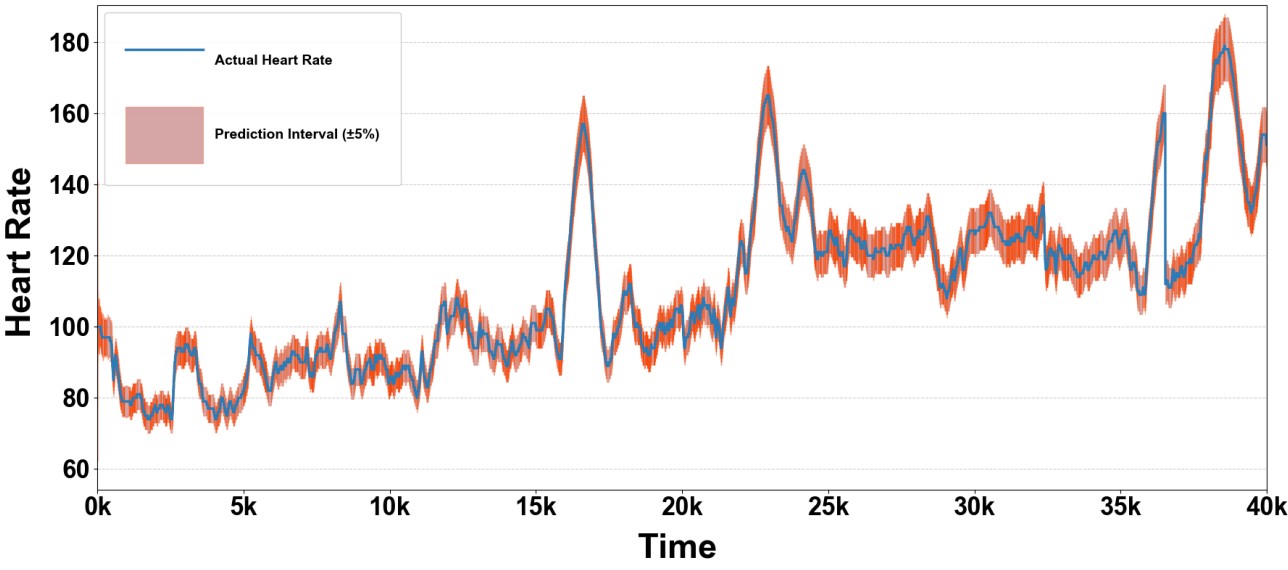

*Figure 21.* To improve visualization, we expanded each prediction interval by ±5 % on both ends (relative to its original length), under the setting of $\mu = 1$. Only intervals that originally covered the true value are shown.

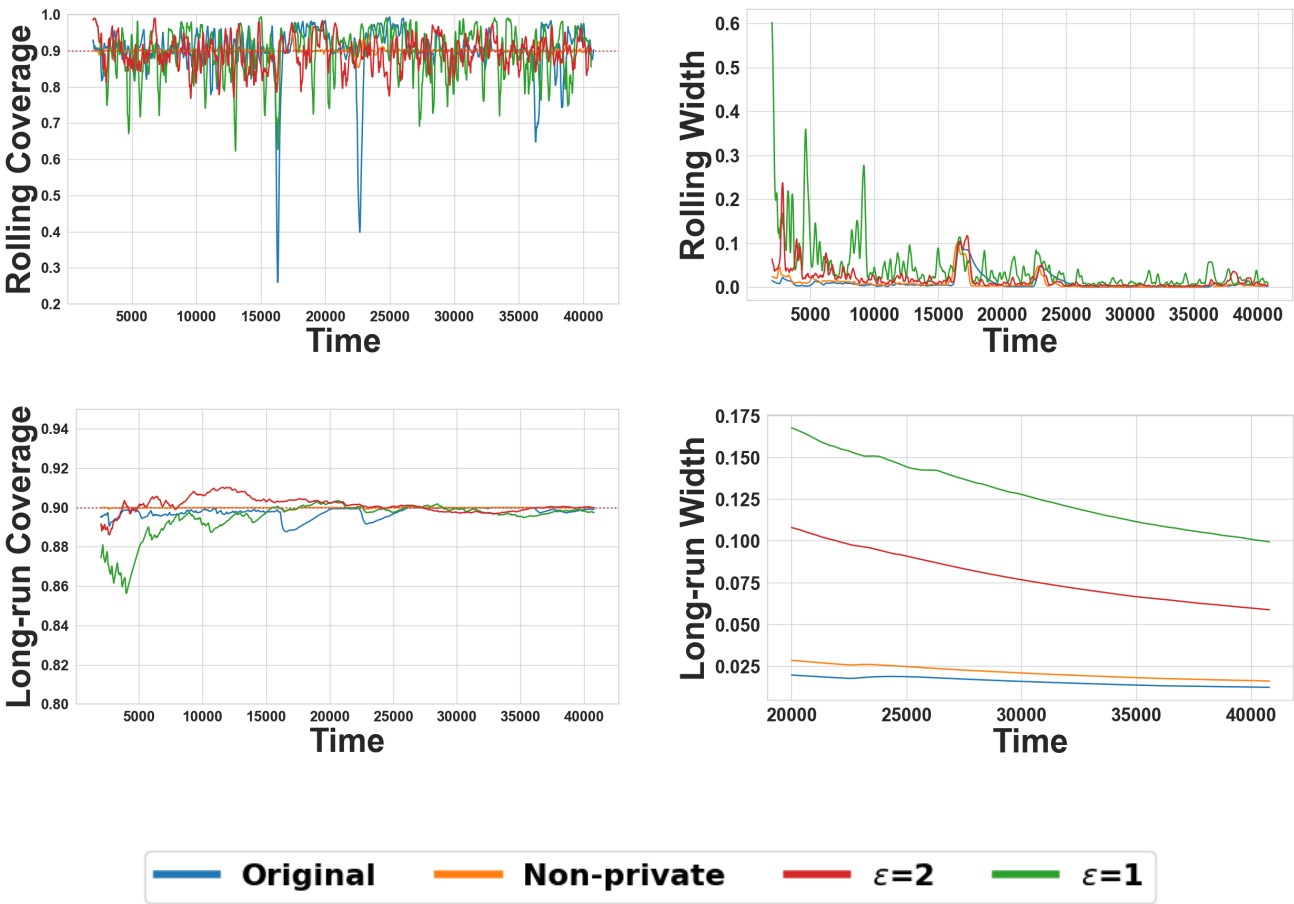

*Figure 22.* **PAMAP2 with $\epsilon$-DP**.The first row shows the rolling coverage and rolling width, averaged over a rolling window of 200 time points. The second row displays the long-run coverage $\sum_{t=1}^{T} \mathbb{I}\{Y_t \in \hat{C}_t(X_t)\}/T$ and the long-run interval width, averaged over 200 repetitions. For stability, the first 2000 time points are excluded from the analysis. The displayed curves are smoothed by taking a rolling average with a window of 50 time points.

