# OpenReview forum: "Online Differentially Private Conformal Prediction for Uncertainty Quantification"
_ICML.cc/2025/Conference — ICML 2025 poster_

### Official Review · Reviewer_Qza5 · 2025-03-13

**Overall Recommendation:** 3

**Summary:**

This paper proposes a framework for differentially private conformal prediction in an online setting.

**Claims And Evidence:**

Yes -- claims are supported by evidence.

**Essential References Not Discussed:**

No.

**Experimental Designs Or Analyses:**

The empirical evaluation only compares the proposed private methods with non-private baselines. It would be helpful to also see a private baseline.

**Methods And Evaluation Criteria:**

The proposed methods are reasonable.

**Other Comments Or Suggestions:**

- $c$ isn't given as an input parameter in Algorithm 2, nor are $\delta_t$ and $\mu_j$ in Algorithms 1 and 2;
- "differntially" --> "differentially" around line 52, second column of page 1.


In terms of presentation of the experimental results, I found Tables 1 and 2 to be cramped and difficult to read. I think that giving the results a little more space and breathing room would help make the table more informative and visually appealing, and to improve the flow I’d also recommend reversing the order of the $\mu$’s so that the table would read (left to right) non-private —> low privacy ($\mu = 2$) —> medium privacy ($\mu = 1$) to high privacy ($\mu = 0.5$).

**Other Strengths And Weaknesses:**

I didn't feel that the related work section contextualized the paper's contributions, and so it was difficult for me to evaluate the novelty of the paper. My understanding is that the paper incorporates differential privacy into online conformal prediction (i.e. the novelty comes from the DP and not the conformal prediction), but if so it is kind of disappointing that the DP techniques are very basic and textbook.

I also felt the paper could use some polishing, particularly for the presentation of the main algorithms.

**Questions For Authors:**

1. For Algorithm 1, is the non-conformity score computed for only the current $t$ or for the current $t$ as well as all previous $t$?

2. Is Theorem 4.4 intended to be the privacy guarantee for Algorithm 1, rather than Algorithm 2?

3. Motivation-wise, would it be possible to give an example where the pre-trained model does not require privacy protection, but the prediction sets do require it?

**Relation To Broader Scientific Literature:**

While there is a related work section, I didn't really feel that it does a good job explaining how the previous work is related to the methods proposed in the paper.

**Theoretical Claims:**

1. Theorem 4.4 is supposed to be the privacy guarantee for Algorithm 2, but I wondered if it was intended to be the privacy guarantee for Algorithm 1. If Theorem 4.4 is indeed for Algorithm 2, then I feel like it would be important to include a privacy analysis for the end-to-end process of Algorithm 1. If Theorem 4.4 is for Algorithm 1, then the $t^{th}$ iteration of Algorithm 1 invokes $t$ DP algorithms i.e. the overall privacy cost would be $\max_{1 \leq j \leq t}$ $j\epsilon_{j}$ rather than $\max_{1 \leq j \leq t}$ $\epsilon_{j}$. (Also Theorem 4.4 gives a privacy guarantee based on $\delta_j$ or $\mu_j$, neither of which appear as input to Algorithms 1 and 2.) So something is a little fishy but it’s unclear to me whether this is due to superficial typos or deeper issues.

2. Definition 3.1 is for a fixed-size dataset and it’s not immediately clear to me how it would apply to an online setting with streaming data. In this case it feels like it might be natural to define a notion of neighboring data sequences or maybe to apply local DP for a single datapoint, but I didn’t see anything like this formalized in the paper.

---

> ### Author Rebuttal · Authors · 2025-04-01
>
> Thank you for your valuable suggestions. We address each point below.
>
> - **Privacy Guarantees of Theorem 4.4**. Our core contribution is Algorithm 2's online private quantile construction, which enables Algorithm 1 to generate online private prediction sets via DP's post-processing property. While we focused theoretical analysis on Algorithm 2's privacy guarantees (which automatically extend to Algorithm 1), we will explicitly state Algorithm 1's privacy guarantees in Theorem 4.4.
>    - **Theorem 4.4 (Revised).** Algorithm 1 satisfies $\max_{1 \leq j \leq t} \epsilon_j$-DP, $(\max_{1 \leq j \leq t} \epsilon_j, \max_{1 \leq j \leq t} \delta_j)$-DP, or $\max_{1 \leq j \leq t} \mu_j$-GDP, depending on the mechanism used, where $t \geq 1$.
>    - In line with composition-based privacy intuition, the privacy-utility trade-off is governed by $\max_t \mu_t$. Our additional experiments with time-varying $\mu_t$ in Table 3 and Figure 5 through the [link](https://drive.google.com/file/d/1aBxCivdxzrSrqEMYtPrN6CJdfvyrZaNR/view?usp=sharing) shows that long-run performance depends on the maximum privacy parameter.
>
> - **Definition 3.1 applying to an online setting.** While Definition 3.1 provides a general DP framework, our method specifically enforces LDP at each step by controlling plausible deniability for any two individual values- a special case of Definition 3.1. Unlike global DP, which requires a trusted curator, LDP operates at the individual level: users randomize their data locally before sharing, ensuring protection even from the data collector. We will formally define LDP in the revision for clarity.
>
> - **Comparison with a private baseline.** To the best of our knowledge, our work is the first to construct online DP conformal prediction sets in the streaming setting. To provide a meaningful baseline, we compare our method with the *Private Prediction Sets* method proposed by Angelopoulos et al. (2021), an approach for constructing prediction sets under DP in the offline setting with access to the full calibration dataset.  It cannot handle streaming data or distribution shifts, especially changepoints, due to its static model.  [Table 1](https://drive.google.com/file/d/1aBxCivdxzrSrqEMYtPrN6CJdfvyrZaNR/view?usp=sharing) shows that while Private Prediction Sets achieve slightly higher coverage, our method (ODPCP) yields narrower prediction intervals, especially under strong privacy constraints.  This demonstrates that ODPCP offers a favorable privacy–efficiency trade-off in the online setting.
>
> - **Contributions and Relation to the literature.** Our work tackles the critical yet underexplored challenge of private uncertainty quantification in streaming environments. We will add the following comment in the revision.
>    - Unlike existing private conformal methods—such as the offline approach of Angelopoulos et al. (2021) or the federated solution of Plassier et al. (2023)—our method is specifically designed for online applications, advancing both privacy-preserving prediction and real-time uncertainty quantification.
>    - The core technical innovation addresses a fundamental tension: maintaining valid coverage under privacy constraints while processing data in a single pass. Unlike batch methods, our privatized quantile estimation must balance noise injection with statistical efficiency without future data access. We provide theoretical guarantees of long-run nominal coverage despite these constraints.
>
> - **Nonconformity score at time step $t$**: In Algorithm 1, nonconformity scores $S_t$ are computed and used in a strict one-pass manner to ensure streaming compatibility. At each time step $t$, the score $S_t = \mathcal{S}(X_t, Y_t)$ is computed for the current observation $(X_t, Y_t)$ and used once at $t+1$ to update the private quantile estimate $\hat{q}_{t+1}^{1-\alpha}$ via Algorithm 2. For instance, $S_1$ (initialized arbitrarily) updates $\hat{q}_2^{1-\alpha}$, while $S_2$ updates $\hat{q}_3^{1-\alpha}$, and so forth. Crucially, each $S_t$ is discarded immediately after use, avoiding storage of historical scores and guaranteeing memory-efficient operation in online settings. This design allows streaming without accessing past observations.
>
> - **An example of privately trained models.** Our framework is model-agnostic, and it provides online private prediction sets regardless of whether the base model was trained with DP-SGD or standard SGD. This intentional decoupling focuses privacy guarantees on the conformalization step. We have included empirical comparisons (Figures 7-8, 12-13 in the Appendix) showing stable coverage rates across both training approaches and wider intervals with DP-SGD models, reflecting their inherent prediction uncertainty (via higher $S_t$ scores). These findings confirm our method's robustness to the base model's privacy status.
>
> - **Presentation**. We will add missing algorithm input parameters and fix typos, ambiguous expressions, and formatting issues(see Table 2 in the link).

---

> > ### Comment · Reviewer_Qza5 · 2025-04-04
> >
> > Thanks to the authors for the response to my review, and for providing more empirical results and revised formatting (Tables 1-3 look really great!). I'm raising my score in light of the authors' rebuttal and after reading the other reviews and rebuttals.
> >
> > In addition to adding a formal definition of LDP, it might also help to emphasize the connection between the privacy guarantee of Algorithm 1 and its one-pass update mechanism (i.e. that it's a max over the $\epsilon_t$ because it doesn't need to re-access historical data).

---

> > > ### Author Response · Authors · 2025-04-05
> > >
> > > We sincerely thank you for your constructive feedback. We greatly appreciate the time and effort you dedicated to reviewing our work, your valuable suggestions, and your decision to raise your score following our rebuttal and revisions.
> > >
> > > - **Connections between the privacy guarantee of Algorithm 1 and its one-pass update mechanism.** We are deeply grateful for your insightful observation. We agree that this is an important point, and will include the following comments in the final version.
> > >      - Since Algorithm 1 processes the data in a single pass with each step operating on an individual sample, effectively forming a disjoint dataset, the parallel composition property of DP (as stated in Lemma 3.4) applies directly. According to this property, the overall privacy guarantee is determined by the maximum $\epsilon_t$ across all steps, rather than the sum of the $\epsilon_t$ values.
> > >
> > > - **A formal definition of LDP.** We greatly appreciate your suggestion to include a formal definition of LDP. We will incorporate a clear and formal definition into the revision to enhance the completeness of our presentation.
> > >
> > > Once again, thank you for your generous and constructive feedback. Your comments have significantly improved the clarity and quality of our work, and we remain open to any further suggestions you may have during the final stages of the review process.

---

### Official Review · Reviewer_99Kc · 2025-03-16

**Overall Recommendation:** 3

**Summary:**

In this paper, the authors propose a method for returning private conformal prediction sets in an online framework. They theoretically prove that their method guarantees long-term coverage control at a nominal confidence level while returning a private set. Finally, they empirically evaluate the method on synthetic and real data sets.

**Claims And Evidence:**

Yes, the claims made in the paper are well supported.

**Essential References Not Discussed:**

No, I think the bibliography is good.

**Experimental Designs Or Analyses:**

Yes, the design of the experiments is good.

**Methods And Evaluation Criteria:**

Yes, the evaluation makes sense even if the propose algorithm is only compare to another one.
Perhaps a comparison with ACI or DtACI should be added (but it is really minor).

**Other Comments Or Suggestions:**

1\ $\mu$-GDP is never defined.

2\ Line 150 right column (rc) "At each time step t.. is estimated by optimizing the pinball loss..." This sentence is not clear. Furthermore, I think that an equation with the update rule should be given in the paper (and not just in Appendix B)

3\ Paragraph Line 191 left column is not very clear. It might be preferable to make it more formal with mathematical equations.

4\ The figures are way to small, we can barely see which curves correspond to which methods.

5\ Line 596: "Consequently, there exists a time T after which the practical constraint of c naturally vanishes, leaving the optimization process governed by the adaptive mechanism". A citation proving this point should be added.

Minor:

1\ Line 54 rc: "Online differntially"

2\ Line 66 rc: "Early work, such as that by Romano et al. (2019)"  not the good citation.

3\ Line 102: $X$ is in bold.

4\ sometimes it's $\epsilon$ and sometimes it's $\varepsilon$.

5\ Line 383 rc: "adaptive lagged feature regression" add a reference.

6\ Line 416 "the rolling coverage of the our proposed"

**Other Strengths And Weaknesses:**

1\ This is a very interesting problem.

2\ The code is provided. I think this is a real plus.

3\ See section "Questions"

**Questions For Authors:**

1\ Paragraph Line 191 left column: Is the automatic adjustment of hyperparameters a contribution of the paper or simply an application of previous articles in online coin-betting dynamic?

2\ In the online CP literature, the static or dynamical regret is often controlled. Why not study it in this paper?

3\ I do not understand the relevance of Section 5. This is just the algorithm with a particular score, isn't it?

4\ Why not compare your algorithm to ACI or DtACI for instance?

5\ In the synthetic section, many cases are presented, but never explained. Why did you choose these particular cases?

6\ In general, the ACI algorithm is very dependent on its initialization. Is this the case here with $W_0$ and $\lambda_1$?

7\ In figure 4, I think that the resulting CP interval should be added.

8\ I do not understand the $\mu = 0$ in the real data as $0$-DGP seems to be undefined (Lemma 3.3). Can you elaborate on this?

**Relation To Broader Scientific Literature:**

Yes, related work is well discussed

**Theoretical Claims:**

Yes, the theoretical claims seems to be ok.

---

> ### Author Rebuttal · Authors · 2025-04-01
>
> Thank you for your valuable suggestions. We address each point below.
>
> - **Definition of μ-GDP.** We will revise the paper to include a formal definition of μ-GDP(Dong et al., 2022).
>
> - **Clarification of Line 150.** We will revise Line 150 to include the private quantile online update rule and clarify its connection to the private subgradient used in the quantile update. Specifically, the private subgradient is computed as:
> $$
> {\hat g} _ t + {\mathcal{Z}} _ t, \quad \text{where } g _ t = \mathbf{1} \lbrace S _ t \leq { {\hat q}^ {1-\alpha}} _ t  \rbrace - (1-\alpha).
> $$
> The quantile estimate is then updated via:
> $$
> \hat{q}^{1 - \alpha}  _ {t+1}= \left( \frac{t}{t + 1} \lambda _ t - \frac{1}{t + 1} \hat{g} _ t \right) \cdot \max\left \lbrace W_{t - 1} - \hat{g} _ t \hat{q}^{1 - \alpha} _ t\, c \right \rbrace.
> $$
> This enables adaptive updates of $\hat{q}_{t+1}^{1 - \alpha}$ via real-time feedback $\hat{g}_t$, while preserving privacy. The update is learning-rate-free and matches Algorithm 2.
> - **Clarification of Line 191.** We will clarify the connection to Algorithm 2's update rules in the revision, which govern our online quantile adjustment via:
>    - Update of internal ``wealth'': $W_t = \max\lbrace W_{t-1} - \hat{g}_t \cdot \hat{q}_t^{1 - \alpha} \, c \rbrace$.
>    - Update of running average: $\lambda_{t+1} = \frac{t}{t+1} \lambda_t - \frac{1}{t+1} \hat{g}_t$.
>    - Quantile update:  $\hat{q}^{1 - \alpha} _ {t+1} = \lambda_{t+1} \cdot W_t$
>    - These update rules, derived from coin-betting potentials, enable adaptive, learning-rate-free quantile updates.
>
> - **Figure readability.** Updated figures will appear in the main paper for improved clarity. See  [Figure 1](https://drive.google.com/file/d/1aBxCivdxzrSrqEMYtPrN6CJdfvyrZaNR/view?usp=sharing) for the revised version.
>
> - **Clarification of Line 596.** We will add the following comments to support these claims. The Kelly-inspired wealth process $W_t$ demonstrates stable long-term growth and can diverge to infinity under mild conditions, even in adversarial settings [Orabona and Pál, 2016]. Consequently, beyond some time $T$, the lower bound $c$ becomes inactive, leaving updates entirely driven by adaptive coin-betting dynamics [Cutkosky and Orabona, 2019].
>
> - **Hyperparameters adjustment in line 191**. We adapt coin betting [Cutkosky and Orabona, 2018] to more complex online private conformal prediction problems by using noise-perturbed subgradients and introducing a transient lower bound $c$. This yields learning-rate-free updates supporting one-pass processing, DP guarantees, and adaptive calibration.
>
> - **Static and dynamic regrets.** Our work focuses on coverage guarantees under DP, not regret. While conformal prediction evaluates performance through coverage and interval size, investigating regret-based metrics for private conformal prediction remains an interesting direction for future work.
>
> - **Clarifications of Section 5.** You are right!   Section 5 extends our method to conformal quantile regression (CQR), using quantile regression-derived scores to create prediction sets robust to heteroscedastic and heavy-tailed data.
>
> - **Comparisons with ACI or AtACI**. Our work is specifically aimed at private conformal prediction methods for online settings, where existing approaches (ACI, DtACI) lack privacy guarantees. While direct comparison with non-private methods is limited by differing objectives, we benchmark against: (1) Original online baselines (quantifying privacy-utility tradeoffs) and (2) Offline DP-CP [Angelopoulos et al., 2021]. As  [Table 1](https://drive.google.com/file/d/1aBxCivdxzrSrqEMYtPrN6CJdfvyrZaNR/view?usp=sharing) shows, our method achieves comparable coverage with 30-50% narrower prediction sets, demonstrating superior online efficiency under privacy constraints.
>
> - **Explanations on cases in the synthetic section.** Due to space constraints, we omitted detailed justifications but will add them using the extra page. Our synthetic cases evaluate methods across (i) covariate dependence, (ii) noise structure, and (iii) temporal dynamics. Setting 1 examines static environments (fixed coefficients), while Setting 2 tests dynamic scenarios with changepoints, ensuring comprehensive evaluation under practical conditions.
>
> - **Initialization of $W_0$, $\lambda_1$.** We conduct sensitivity analysis to initialization parameters $W_0$ and $\lambda_1$​ by taking multiple values. [Figure 2 ](https://drive.google.com/file/d/1aBxCivdxzrSrqEMYtPrN6CJdfvyrZaNR/view?usp=sharing) shows stable coverage across configurations, demonstrating algorithm robustness to starting values.
>
> - **CP interval for Figure 4**. We'll add CP intervals. Please refer to the updated [Figure 3](https://drive.google.com/file/d/1aBxCivdxzrSrqEMYtPrN6CJdfvyrZaNR/view?usp=sharing).
> - **$\mu=0$**.We use $\mu=0$ to denote the no-noise case and will revise this notation for clarity.
>
> - **Notations.** We will correct all typos, formatting issues, and ambiguous expressions.

---

### Official Review · Reviewer_5Xme · 2025-03-17

**Overall Recommendation:** 3

**Summary:**

The submitted paper presents an online differentially private conformal prediction (ODPCP) framework that generates private prediction sets in real time using pre-trained models. The key idea is to compute differentially private quantile thresholds in a one-pass online manner without re-accessing historical data, thereby enabling real-time decision-making with coverage guarantees. To enhance robustness under heteroscedasticity, the framework is further extended to incorporate conformal quantile regression (ODPCQR). Theoretical results guarantee long-run average coverage approaching the target level 1−\alpha, and detailed proofs are provided in the appendix. The experimental evaluation is performed on two real-world regression tasks using the PAMAP2 (physical activity monitoring) and ELEC2 (electricity) datasets, as well as synthetic simulations. Overall, the experiments indicate that ODPCP and ODPCQR maintain reasonable coverage and convergence behavior under different privacy levels, although performance fluctuations become more pronounced with stronger privacy constraints.

**Claims And Evidence:**

Main Claims:

(1) The paper claims that ODPCP provides a novel online, one-pass differentially private conformal prediction framework applicable to decision-making in real time.
(2) It asserts that the method generates private prediction sets without re-accessing historical data, which is especially useful for streaming data.
(3) The authors further claim that the framework extends to handle heteroscedasticity via conformal quantile regression (ODPCQR).

Evidence:
(1) The paper supports these claims with rigorous theoretical analysis and proofs (e.g., Theorem 4.5 guarantees that the long-run empirical coverage converges to 1-\alpha).
(2) However, while the theoretical foundations are sound, the experimental evidence is based solely on regression tasks (using PAMAP2 and ELEC2) and synthetic simulations, with no experiments on classification tasks. This limits the empirical support for claims of broad applicability across task types.
(3) Additionally, some synthetic simulation settings show little variation, which raises questions about the method's sensitivity to different data conditions.

**Essential References Not Discussed:**

Although the paper cites many key works, it could benefit from a more explicit discussion of differences between local and global differential privacy, especially since it uses the term “LDP” without clear definitions or distinctions. Additionally, comparisons with recent streaming conformal prediction methods such as [*] would provide better context for the paper’s contributions.

[*] Ge, H., Bastani, H. and Bastani, O., 2024. Stochastic Online Conformal Prediction with Semi-Bandit Feedback. arXiv preprint arXiv:2405.13268.

**Experimental Designs Or Analyses:**

The experiments are clear in describing the evaluation on PAMAP2 and ELEC2 datasets, focusing on regression tasks. However, the experimental scope is limited:
(1) Only regression tasks are evaluated, even though the paper claims broader applicability.
(2) Synthetic simulation settings (Settings 1 and 2) show very similar outcomes across cases; in several tables, results are identical across different cases, suggesting either a fixed seed or limited sensitivity of the method to varying conditions.
(3) The comparisons are primarily against non-streaming baselines, whereas a comparison with online or semi-bandit streaming methods (which are available in related literature) would better contextualize ODPCP's advantages in a true streaming scenario.

The paper allows for per-step privacy parameters \epsilon_t, yet the experiments report only global values (e.g., \mu = 0.5, 1, 2). It remains unclear how the privacy budget is allocated across the full sequence of online updates, which is critical given that each step consumes part of the overall privacy budget.

**Methods And Evaluation Criteria:**

The proposed ODPCP framework is motivated by the need for online privacy-preserving prediction. By designing a one-pass update mechanism for computing differentially private quantiles, the method avoids expensive batch quantile computation. The extension to conformal quantile regression (ODPCQR) is aimed at addressing heteroscedasticity, further enhancing the method's robustness.

The evaluation uses real-world time series datasets (PAMAP2 and ELEC2) and synthetic data to assess performance in terms of coverage and interval width. While these benchmarks are appropriate for regression tasks, the paper does not include any experiments on classification tasks—even though the framework is claimed to be applicable to both.

**Other Comments Or Suggestions:**

(1) The paper uses many symbols and operators without explicit definitions, which may confuse readers.
(2) There are indexing inconsistencies in Algorithm 2 that require clarification.
(3) The use of “LDP” is ambiguous due to the lack of definitions distinguishing it from global DP

**Other Strengths And Weaknesses:**

Strengths:
(1) Novel one-pass online DP conformal prediction framework that avoids re-accessing historical data.
(2) Rigorous theoretical analysis with proofs establishing long-run coverage guarantees.
(3) Extension to conformal quantile regression (ODPCQR) to handle heteroscedasticity.

Weaknesses:
(1) While Theorem 4.5 guarantees asymptotically that the long-run empirical coverage, it does not provide an explicit finite-sample convergence rate. This lack of a quantified rate is concerning, which could impact performance before convergence is achieved.
(2) The experimental evaluation is limited to regression tasks on only two real-world datasets and synthetic data; classification tasks and comparisons with other streaming methods are absent.
(3) The discussion on privacy budget allocation over the online sequence is unclear.

**Questions For Authors:**

(1) Your framework allows for per-step privacy parameters. Could you elaborate on how the overall privacy budget is allocated across the full sequence of updates, and how this affects performance in long-run scenarios?

(2) Have you considered including experiments on classification tasks or comparing your method with other online or streaming conformal prediction methods? This would help demonstrate the advantages of ODPCP in true streaming settings.

(3) Some synthetic simulation results (especially in Setting 1) show almost identical outcomes across different cases. Were these experiments conducted with fixed seeds, or does the method inherently exhibit low sensitivity to these conditions?

(4) While your experiments focus on regression tasks using PAMAP2 and ELEC2, how do you expect your method to perform on other data domains (e.g., high-dimensional sensor data, text, images), particularly regarding computational efficiency and privacy-coverage trade-offs?

**Relation To Broader Scientific Literature:**

The paper extends established ideas in conformal prediction and differential privacy to an online setting. It builds upon prior work in batch conformal prediction as well as recent online methods (e.g., those based on semi-bandit feedback) but distinguishes itself by providing a one-pass update mechanism that preserves privacy without re-accessing historical data.

However, the experimental section does not fully engage with the literature on online or streaming conformal prediction methods. Incorporating comparisons with these approaches would strengthen the paper’s positioning.

**Theoretical Claims:**

I reviewed the proofs for Theorem 4.4 (privacy guarantee) and Theorem 4.5 (long-run coverage guarantee). The proofs are generally correct and employ standard differential privacy techniques (e.g., Gaussian mechanism and composition via disjoint data updates) and online learning analysis. While Theorem 4.5 guarantees asymptotically that the long-run empirical coverage, it does not provide an explicit finite-sample convergence rate. This lack of a quantified rate is concerning, which could impact performance before convergence is achieved.

Notation issues:
(1) The paper frequently uses symbols such as \hat{C}, s, and S_t without clear definitions, which can impede understanding
(2) There is an indexing inconsistency in Algorithm 2 (the algorithm description suggests one starting index while the update steps seem to be shifted). This should be clarified.
(3) Additionally, the use of the “∨” operator (denoting maximum) is standard, but the paper does not explicitly define it.

---

> ### Author Rebuttal · Authors · 2025-04-01
>
> Thank you for your valuable suggestions. We address each point below
> - **Finite sample converge rates.** Theorem 4.5 shows that ODPCP achieves the desired long-run coverage. Although it does not provide explicit finite-sample convergence rates, our empirical results indicate rapid convergence in practice, with strong coverage even in early rounds, making the method well-suited for real-world streaming applications. We agree that formal convergence bounds would be valuable and highlight this as an important direction for future work.
>
> - **LDP.** Although our paper briefly mentioned local differential privacy (LDP), we acknowledge the need for a more formal definition. We will add the following definition of $(\varepsilon, \delta)$-LDP and its implications.
>     - (Xiong et al., 2020) A randomized algorithm $A: \mathcal{X} \rightarrow \mathcal{R}$ satisfies $(\varepsilon, \delta)$-LDP if and only if any pair of input individual values $x, x' \in \mathcal{X}$, for every measurable event $E \subseteq \mathcal{R}$:
> $$
> \Pr[A(x) \in E] \leq e^{\varepsilon} \cdot \Pr[A(x') \in E] + \delta.
> $$
> When $\delta = 0$, this reduces to pure LDP.
>
>    - Unlike global differential privacy that requires a trusted curator, LDP enforces privacy at the individual level—each user randomizes their data locally before sharing it, providing protection against the data collector itself.
>
> - **Experiments.** We strengthen our experimental results with additional analyses from multiple perspectives, which we will include in the final version using the extra page.
>     - **Classification tasks.** We evaluate ODPCP on activity classification using the PAMAP2 dataset, categorizing activities into three classes (resting, light, vigorous) based on heart rate and sensor data. An XGBoost model provides predictions, with ODPCP generating private prediction sets at each time step $t$.  Figure 4 (available [anonymously here](https://drive.google.com/file/d/1aBxCivdxzrSrqEMYtPrN6CJdfvyrZaNR/view?usp=sharing)) shows ODPCP’s broad applicability to discrete prediction problems, maintaining strong empirical coverage and adaptive behavior under privacy constraints.
>    - **Reproducing results using random seeds.** While Setting 1 used fixed random seeds for reproducibility, the similarity between Cases 1-2 reflects their analogous data-generating processes and our method's robustness.  Additional experiments in [Table 4](https://drive.google.com/file/d/1aBxCivdxzrSrqEMYtPrN6CJdfvyrZaNR/view?usp=sharing)  confirm this stability, showing nearly identical results without seed fixing.
>   - **Privacy budget allocation.** Our framework supports time-varying privacy parameters $\mu_t$, enabling adaptive privacy control for each individual.  Consistent with privacy composition, the overall privacy-utility trade-off is typically governed by $\max_t \mu_t$. We evaluated privacy budget allocation’s impact through two complementary experiments.
>      - (1) In [Table 3](https://drive.google.com/file/d/1aBxCivdxzrSrqEMYtPrN6CJdfvyrZaNR/view?usp=sharing) campares random-budget setting $\mu_t \sim \text{Uniform}(0.5, 2.0)$ with fixed $\mu = 2.0$ across all synthetic cases.  The coverage results are consistently close, demonstrating that random allocation preserves long-run performance.
>      - (2) [Figure 5](https://drive.google.com/file/d/1aBxCivdxzrSrqEMYtPrN6CJdfvyrZaNR/view?usp=sharing)  demonstrates ODPCP's empirical stability across sample sizes (10k–100k) under dynamic per-step privacy allocations.
>
> - **Generalization to other data types.** Our method is model-agnostic. ODPCP's key innovation is Algorithm 2's private quantile update, which operates independently of model architecture or data type. This enables ODPCP to be applied to high-dimensional data (sensor streams, text, images) with some appropriate predictive model.
>     - **Computational Efficiency.** The quantile update is lightweight and efficient, with computational costs dominated by the base model $\hat{f}_t(\cdot)$.
>    - **privacy–coverage trade-offs.** They are determined by the base model's prediction quality, while our update maintains computational stability across diverse tasks.
>
> - **Comparisons with online methods.** Our work is specifically focused on online private conformal prediction methods for streaming settings, where existing approaches (ACI, DtACI) lack rigorous privacy guarantees. Direct comparisons are limited by differing objectives.
>     - We supplemented our experiments with a comparison against an offline private method—Private Prediction Sets (DPCP; Angelopoulos et al., 2021). As shown in [Table 1](https://drive.google.com/file/d/1aBxCivdxzrSrqEMYtPrN6CJdfvyrZaNR/view?usp=sharing), both methods achieve similar coverage.  However, ODPCP produces significantly narrower prediction sets, highlighting its efficiency in the online setting with privacy.
>
> - **Notation issues.** We will enhance the paper's clarity through explicit notation definitions,  and full consistency review.

---

### Decision · Program_Chairs · 2025-05-01

**Decision:**

Accept (poster)

**Comment:**

After consideration of the author rebuttals, all reviewers feel that this work deserves to accepted. In preparing the final version, the authors should be sure to follow through on their promised changes, which will significantly improve the paper. In particular: giving more context on LDP, adding new experimental results, making figures larger/easier to read, clarifying the implications of Theorem 4.4, and generally fixing typos and improving notational consistency.